# Augmentations in Hypergraph Contrastive Learning: Fabricated and Generative

**Tianxin Wei**[1*], **Yuning You**[2*], **Tianlong Chen**[3], **Yang Shen**[2], **Jingrui He**[1], **Zhangyang Wang**[3]
[1]University of Illinois Urbana-Champaign, [2]Texas A&M University, [3]University of Texas at Austin
{twei10,jingrui}@illinois.edu, {yuning.you,yshen}@tamu.edu,
{tianlong.chen,atlaswang}@utexas.edu

## Abstract

This paper targets at improving the generalizability of hypergraph neural networks in the low-label regime, through applying the contrastive learning approach from images/graphs (we refer to it as **HyperGCL**). We focus on the following question: *How to construct contrastive views for hypergraphs via augmentations?* We provide the solutions in two folds. First, guided by domain knowledge, we **fabricate** two schemes to augment hyperedges with higher-order relations encoded, and adopt three vertex augmentation strategies from graph-structured data. Second, in search of more effective views in a data-driven manner, we for the first time propose a hypergraph generative model to **generate** augmented views, and then an end-to-end differentiable pipeline to jointly learn hypergraph augmentations and model parameters. Our technical innovations are reflected in designing both fabricated and generative augmentations of hypergraphs. The experimental findings include: (i) Among fabricated augmentations in HyperGCL, augmenting hyperedges provides the most numerical gains, implying that higher-order information in structures is usually more downstream-relevant; (ii) Generative augmentations do better in preserving higher-order information to further benefit generalizability; (iii) HyperGCL also boosts robustness and fairness in hypergraph representation learning. Codes are released at `https://github.com/weitianxin/HyperGCL`.

## 1 Introduction

Hypergraphs have raised a surge of interests in the research community [1, 2, 3] due to their innate capability of capturing higher-order relations [4]. They offer a powerful tool to model complicated topological structures in broad applications, e.g., recommender systems [5, 6], financial analyses [7, 8], and bioinformatics [9, 8, 10]. Concomitant with the trend, hypergraph neural networks (HyperGNNs) have recently been developed [1, 2, 3] for hypergraph representation learning.

This paper focuses on the few-shot scenarios of hypergraphs, i.e., task-specific labels are scarce, which are ubiquitous in real-world applications of hypergraphs [5, 7, 9] and empirically restrict the generalizability of HyperGNNs. Inspired by the emerging self-supervised learning on images/graphs [11, 12, 13, 14, 15, 16], especially the contrastive approaches [12, 14, 17, 18, 19, 20, 21, 22, 23, 24, 25], we set out to leverage contrastive self-supervision to address the problem.

Nevertheless, one challenge stands out: *How to build contrastive views for hypergraphs?* The success of contrastive learning hinges on the appropriate view construction, otherwise it would result in "negative transfer" [12, 14]. However, it is non-trivial to build hypergraph views due to their overly intricate topology, i.e., there are $\sum_{e=1}^{N} \binom{N}{e}$ possibilities for one hyperedge on $N$ vertices, versus $\binom{N}{2}$ for one edge in graphs. To date, the only way of contrasting is between the representations

---

*Equal contribution.

of hypergraphs and their clique-expansion graphs [26, 27], which is computationally expensive as multiple neural networks of different modalities (hypergraphs and variants of expanded graphs) need to be optimized. More importantly, contrasting on clique expansion has the risk of losing higher-order information via pulling representations of hypergraphs and graphs close.

**Contributions.** Motivated by [12, 14] that appropriate data augmentations suffice for the effective contrastive views, and intuitively they are more capable of preserving higher-order relations in hypergraphs compared to clique expansion, we explore on the question in this paper, how to design augmented views of hypergraphs in contrastive learning (**HyperGCL**). Our answers are in two folds.

We first assay whether **fabricated** augmentations guided by domain knowledge are suited for HyperGCL. Since hypergraphs are composed of hyperedges and vertices, to augment hyperedges, we propose two strategies that (i) directly perturb on hyperedges, and (ii) perturb on the "edges" between hyperedges and vertices in the converted bipartite graph; To augment vertices, we adopt three schemes of vertex dropping, attribute masking and subgraph from graph-structured data [14]. Our finding is that, different from the fact that vertex augmentations benefit more on graphs, *hypergraphs mostly benefit from hyperedge augmentations* (up to 9% improvement), revealing that higher-order information encoded in hyperedges is usually more downstream-relevant (than information in vertices).

Furthermore, in search of even better augmented views but in a data-driven manner, we study whether/how augmentations of hypergraphs could be learned during contrastive learning. To this end, for the first time, we propose a novel variational hypergraph auto-encoder architecture, as a hypergraph **generative** model, to parameterize a certain augmentation space of hypergraphs. In addition, we propose an end-to-end differentiable pipeline utilizing Gumbel-Softmax [28], to jointly learn hypergraph augmentations and model parameters. Our observation is that generative augmentations can better capture the higher-order information and achieve state-of-the-art performance on most of the benchmark data sets (up to 20% improvement).

The aforementioned empirical evidences (for *generalizability*) are drawn from comprehensive experiments on 13 datasets. Moreover, we introduce the *robustness* and *fairness* evaluation for hypergraphs, and show that HyperGCL in addition boosts robustness against adversarial attacks and imposes fairness with regard to sensitive attributes.

The rest of the paper is organized as follows. We discuss the related work in Section 2, introduce HyperGCL in Section 3, present the experimental results in Section 4, and conclude in Section 5.

## 2 Related Work

**Hypergraph neural networks.** Hypergraphs, which are able to encode higher-order relationships, have attracted significant attentions in recent years. In the machine learning community, hypergraph neural networks are developed for effective hypergraph representations. HGNN [1] adopt the clique expansion technique and designs the weighted hypergraph Laplacian for message passing. HyperGCN [2] proposes the generalized hypergraph Laplacian and explores adding the hyperedge information through mediators. The attention mechanism [29, 30] is also designed to learn the importance within hypergraphs. However, the expanded graph will inevitably cause distortion and lead to unsatisfactory performance. There is also another line of works such as UniGNN [31] and HyperSAGE [32] which try to perform message passing directly on the hypergraph to avoid the information loss. A recent work [3] provides an AllSet framework to unify the existing studies with high expressive power and achieves state-of-the-art performance on comprehensive benchmarks. The work utilizes deep multiset functions [33] to identify the propagation and aggregation rules in a data-driven manner.

**Contrastive self-supervised learning.** Contrastive self-supervision [12, 34, 35] has achieved unprecedented success in computer vision. The core idea is to learn an embedding space where samples from the same instance are pulled closer and samples from different instances are pushed apart. Recent works start to cross-pollinate between contrastive learning and graph neural networks to for more generalizable graph representations. Typically, they design some fabricated augmentations guided by domain knowledge, such as edge perturbation, feature masking or vertex dropping, etc. Nevertheless, contrastive learning on hypergraphs remains largely unexplored. Most existing works [6, 36, 26, 37] design pretext tasks for hypergraphs and mainly focus on recommender systems [38, 39, 40, 41], via contrasting between graphs and hypergraphs which might lose important higher-order information. In this work, we explore on the structure of hypergraph itself to construct contrastive views.

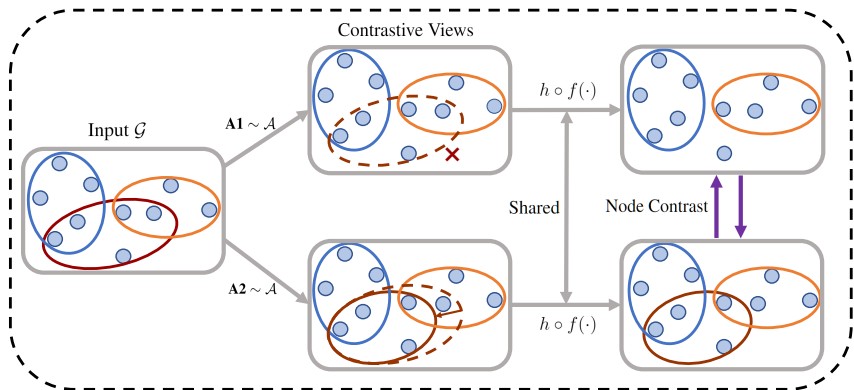

Figure 1: The framework of hypergraph contrastive learning (HyperGCL). The ellipses represent the hyperedges. Two contrastive views are generated by hypergraph augmentations **A1** and **A2** from the augmentation collection $\mathcal{A}$. $f(\cdot)$ and $h(\cdot)$ are shared encoder and projection head respectively. In the figure, we show two examples of hypergraph augmentations. At the top, the dotted ellipse denotes the deleted hyperedge. At the bottom, one vertex in the dotted hyperedge is removed.

# 3 Methods

## 3.1 Hypergraph Contrastive Learning

A hypergraph is denoted as $\mathcal{G} = \{\mathcal{V}, \mathcal{E}\} \in \mathbb{G}$ where $\mathcal{V} = \{v_1, ..., v_{|\mathcal{V}|}\}$ is the set of vertices and $\mathcal{E} = \{e_1, ..., e_{|\mathcal{E}|}\}$ is the set of hyperedges. Each hyperedge $e_n = \{v_1, ..., v_{|e_n|}\}$ represents the higher-order interaction among a set of vertices. State-of-the-art approaches to encode such complex structures are hypergraph neural networks (HyperGNNs) [1, 2, 3], mapping the hypergraph to a $D$-dimension latent space via $f : \mathbb{G} \to \mathbb{R}^D$ with higher-order message passing.

Motivated from learning on images/graphs, we adopt contrastive learning to further improve the generalizability of HyperGNNs in the low-label regime (HyperGCL). Main components of our HyperGCL, similar to images/graphs [12, 14] include: (i) **hypergraph augmentations for contrastive views**, (ii) HyperGNNs as hypergraph encoders, (iii) projection head $h(\cdot)$ for representations, and (iv) contrastive loss for optimization. The overall pipeline is shown in Figure 1. Detailed descriptions and training procedure are shown in Appendix B. The main challenge here is how to effectively augment hypergraphs to build contrastive views.

## 3.2 Fabricated Augmentations for Hypergraphs

We first explore whether manually designed augmentations are suited for Hyper-GCL. Since hyperedges and vertices compose a hypergraph, augmentations are fabricated with regards to topology and node features, respectively.

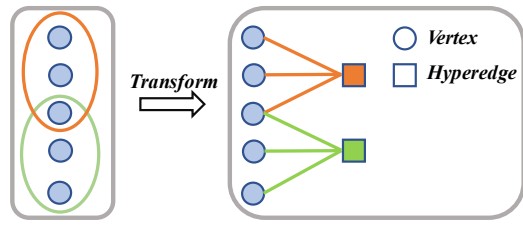

Figure 2: Conversion from hypergraph to equivalent bipartite graph.

**A1. Perturbing hyperedges.** The most direct augmentation on higher-order interactions is to perturb on the set of hyperedges. Since adding a hyperedge is confronted with the combinatorial challenge (see Sec. 1 of introduction), here we focus on randomly removing the existing hyperedges following an i.i.d. Bernoulli distribution. The underlying assumption is that the partially missing higher-order relations do not significantly affect the semantic meaning of hypergraphs.

**A2. Perturbing edges in equivalent bipartite graph.** To augment higher-order relations in a more fine-grained way, we first convert the hypergraph into the equivalent bipartite graph, where two disjoint sets of vertices represent vertices and hyperedges in the hypergraph, respectively (see Figure 2). On top of the bipartite graph, we perform random removal of edges. A2 disrupts the higher-order relations via randomly kicking out vertices from hyperedges, enforcing the semantics of hypergraph representations to be robust to such disruption. A2 is essentially the generalized version of A1.

Moreover, we find that vertex augmentations for graph-structured data [14] are applicable to hypergraphs. Therefore, we adopt three additional schemes of vertex dropping (**A3**), attribute masking (**A4**) and subgraph (**A5**) into our experiments, with similar prior knowledge incorporated as in [14].

## 3.3 Generative Models for Hypergraph Augmentations

Manually designing augmentation operators requires a wealth of domain knowledge, and might lead to sub-optimal solutions even with extensive trial-and-errors. We next study whether/how augmentations of hypergraphs could be learned during contrastive learning. Two questions need to be answered here: (i) How to parameterize the augmentation space of hypergraphs? (ii) How to incorporate the learnable augmentations into contrastive learning?

### 3.3.1 Hypergraph Generative Models for Augmentations

Considering an augmentation operator defined as the stochastic mapping between two hypergraph manifolds that $g : \mathbb{G} \to \mathbb{G}$, a natural thought is to adopt the generative model to parameterize the augmentation space, which in general is composed of a deterministic encoder $h_1 : \mathbb{G} \to \mathbb{R}^{D'}$ and a stochastic decoder (or sampler) $h_2 : \mathbb{R}^{D'} \to \mathbb{G}$. In this way, $g = h_1 \circ h_2$.

Following this thought, inspired by the well-studied generative models with variational inference [42, 43], we propose a novel variational hypergraph auto-encoder architecture (VHGAE). To the best of our knowledge, this is the first hypergraph generative model for generating augmentations of hypergraphs, which will be used as **A6**. Notice that here it only parametrizes the augmentation space of edge perturbation, and in the future node perturbation would be included. VHGAE consists of the encoder and decoder neural networks. The overall framework is shown in Figure 3.

**Encoder.** The encoder embeds hypergraphs into latent representations. Instead of embedding a hypergraph into a single vector, we follow VGAE [43] to embed it into a set of vertex and in additional hyperedge representations, to facilitate the further decoding process of non-Euclidean structures. We adopt two HyperGNNs, $h_1^\mu$ and $h_1^\sigma$, to encode the mean and the logarithmic standard deviation for variational distributions of vertex and hyperedge representations $z_\mathcal{V} \sim q_\phi(z_\mathcal{V}|\mathcal{G}) = \mathcal{N}(\mu_\mathcal{V}, \sigma_\mathcal{V}^2), z_\mathcal{E} \sim q_\phi(z_\mathcal{E}|\mathcal{G}) = \mathcal{N}(\mu_\mathcal{E}, \sigma_\mathcal{E}^2)$ as follows (please refer to Appendix B for the detailed computing pipeline):

$$\mu_\mathcal{V}, \mu_\mathcal{E} = h_1^\mu(\mathcal{G}), \qquad \log(\sigma_\mathcal{V}), \log(\sigma_\mathcal{E}) = h_1^\sigma(\mathcal{G}), \tag{1}$$

where $\mu \in \mathbb{R}^{D' \times |\mathcal{V}|}, \log(\sigma) \in \mathbb{R}^{D' \times |\mathcal{V}|}$. We here leverage the higher-order message passing in HyperGNNs for a better encoding capability.

**Decoder.** With the learned vertex and hyperedge variational distributions, the decoder attempts to reconstruct the higher-order relations of hypergraphs. However, modeling the space of higher-order interactions encounters the combinatorial challenge (see Section 1). Adopting the similar strategy as in the augmentation A2 (see Section 3.2), we designate the decoder to recover the relations on the converted bipartite graph $\tilde{\mathcal{G}} = \{\tilde{\mathcal{V}}, \tilde{\mathcal{E}}\}$ for approximation. Mathematically, we formulate decoding as:

$$p(\mathcal{G}|z_\mathcal{V}, z_\mathcal{E}) \approx p(\tilde{\mathcal{G}}|z_\mathcal{V}, z_\mathcal{E}) = \prod_{e=1}^{|\mathcal{E}|} \prod_{v=1}^{|\mathcal{V}|} p(\tilde{\mathcal{E}}_{v,e}|z_v, z_e) = \prod_{e=1}^{|\mathcal{E}|} \prod_{v=1}^{|\mathcal{V}|} \text{Sigmoid}(z_v^T z_e), \tag{2}$$

where $w_{ve} = z_v^T z_e$ is the learned edge logit. On the decoded topological distribution of the bipartite graph, we perform sampling and then convert the sample back to the hypergraph (the conversion between hypergraphs and bipartite graphs is lossless).

**Generator optimization.** With variational inference [42, 44, 45], we optimize the hypergraph generator on the evidence lower bound (ELBO) as follows:

$$\text{ELBO} = \mathbb{E}_{q_\phi(z_\mathcal{E}|\mathcal{G})} \mathbb{E}_{q_\phi(z_\mathcal{V}|\mathcal{G})} \left[\log p_\theta(\mathcal{G}|z_v, z_e)\right] - \text{KL}[q_\phi(z_\mathcal{V} \mid \mathcal{G}) \mid p(z_\mathcal{V})] - \text{KL}[q_\phi(z_\mathcal{E} \mid \mathcal{G}) \mid p(z_\mathcal{E})], \tag{3}$$

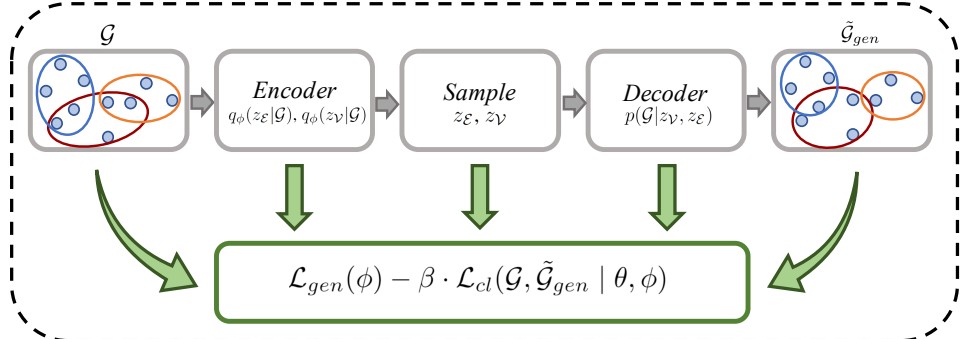

Figure 3: Framework of the proposed variational hypergraph auto-encoder (VHGAE). The green lines indicate these modules participated in the optimization process.

where $q_\phi(z_\mathcal{E}|\mathcal{G})$ and $q_\phi(z_\mathcal{V}|\mathcal{G})$ are their variational distribution, $p(z_\mathcal{V})$ and $p(z_\mathcal{E})$ are default Gaussian priors with $p(z_\mathcal{V}) \sim \mathcal{N}(0, I)$, $p(z_\mathcal{E}) \sim \mathcal{N}(0, I)$. When generating hypergraphs, the generator would sample the relations on the converted bipartite graph with probability $p(\tilde{\mathcal{G}}|z_\mathcal{V}, z_\mathcal{E})$.

### 3.3.2 Jointly Augmenting and Contrasting with Gumbel-Softmax

With hypergraph augmentations parametrized with generative models, the next step is to incorporate augmentation learning into HyperGCL. The main barrier results from the discrete sampling of hyperedges which is non-differentiable. To tackle it, we leverage the Gumbel-Softmax trick [28] for the hyperedge distribution as:

$$
\begin{aligned}
T(\mathcal{G}) &= \text{Gumbel-Softmax}(p(\mathcal{G} \mid z_\mathcal{V}, z_\mathcal{E})) \\
&= \text{Sigmoid}((w_{\mathcal{V}\mathcal{E}} + \log(\delta) - \log(1 - \delta))/\tau) \\
\tilde{\mathcal{G}}_{gen} &= T(\mathcal{G}) \circ \mathcal{G},
\end{aligned}
\tag{4}
$$

where $w_{\mathcal{V}\mathcal{E}}$ denotes the learned edge logits (before Sigmoid) and $\delta \sim \text{Uniform}(0, 1)$. When hyperparameter temperature $\tau \to 0$, the results get closer to being binary. $T$ is the sampled one-hot vector for each hyperedge-vertex interaction in the hypergraph $\mathcal{G}$. Then the sampled vector will be applied to perform augmentation. During the Gumbel-Softmax, we leverage the reparametrization trick [42] to smooth the gradient and make the sample operation differentiable. Thus, this objective can be optimized in an end-to-end manner as:

$$
\min_\phi \mathcal{L}_{gen}(\phi) - \beta \cdot \mathcal{L}_{cl}(\mathcal{G}, \tilde{\mathcal{G}}_{gen} \mid \theta, \phi),
\tag{5}
$$

where $\mathcal{L}_{gen} = -\text{ELBO}$ is the generator loss to be minimized, $\beta$ is the tradeoff factor. Due to the computational cost of collaboratively optimizing two generative views, we train one VHGAE to produce one generative view, with the other view $\tilde{\mathcal{G}}_p$ is kept as fabricated. To be specific, (i) independently optimizing two hypergraph generators is of reasonable budgets but would lead to distribution collapse (i.e., two hypergraph generators output the same distribution) [1,2] which results in less effective generative views, while (ii) the collaborative optimization techniques for graph generators (e.g. REINFORCE on the rewards of generative graph structures) are not directly applicable to HyperGCL due to the combinatorial challenge of hypergraph structures (which is computationally expensive). The goal of this multi-task loss is to generate stronger augmentation (maximize contrastive loss) to push HyperGNN to avoid capturing redundant information during the representation learning, while at the same time learning the hypergraph data distribution.

## 4 Experiments

### 4.1 Setup

We examine our methods on the most comprehensive hypergraph benchmarks with 13 data sets, with statistics shown in Table 1. Please refer to Appendix C for detailed information. We focus on semi-supervised vertex classification in the transductive setting. Different from the existing work

Table 1: Data statistics: $h_e$, $h_v$ are the node homophily and hyperedge homophily in hypergraph. Higher value indicate the hypergraph is more homogeneous. Details can be found in Appendix C.

| | Cora | Citeseer | Pubmed | Cora-CA | DBLP-CA | Zoo | 20News | Mushroom | NTU2012 | ModelNet40 | Yelp | House | Walmart |
|---|---|---|---|---|---|---|---|---|---|---|---|---|---|
| $|\mathcal{V}|$ | 2708 | 3312 | 19717 | 2708 | 41302 | 101 | 16242 | 8124 | 2012 | 12311 | 50758 | 1290 | 88860 |
| $|\mathcal{E}|$ | 1579 | 1079 | 7963 | 1072 | 22363 | 43 | 100 | 298 | 2012 | 12311 | 679302 | 341 | 69906 |
| # feature | 1433 | 3703 | 500 | 1433 | 1425 | 16 | 100 | 22 | 100 | 100 | 1862 | 100 | 100 |
| # class | 7 | 6 | 3 | 7 | 6 | 7 | 4 | 2 | 67 | 40 | 9 | 2 | 11 |
| $h_e$ | 0.86 | 0.83 | 0.88 | 0.88 | 0.93 | 0.66 | 0.73 | 0.96 | 0.87 | 0.92 | 0.57 | 0.58 | 0.75 |
| $h_v$ | 0.84 | 0.78 | 0.79 | 0.79 | 0.88 | 0.35 | 0.49 | 0.87 | 0.81 | 0.88 | 0.26 | 0.52 | 0.55 |

[3] that leverages 50% of all vertexes as the training set, we focus on the more low-label regime of challenging and practical applications. By default, we split the data into training/validation/test samples using (10%/10%/80%) splitting percentages. Each experiment is run for 20 different data splits and initialization with mean and standard deviation reported. We adopt state-of-the-art SetGNN [3] as the backbone HyperGNN architecture. For baselines, we compare two existing hypergraph self-supervised approaches [36] and [26] in recommender systems, denoted as Self and Con. They conduct self-supervised learning between the hypergraph and conventional graph. By default, we adopt multi-task training to incorporate contrastive self-supervision because it performs the best as shown in the comparison in Section 4.2. All the implementation details are listed in Appendix C. More experiments of the hyperparameters study are given in Appendix A.

## 4.2 Results

**Comparison among different hypergraph augmentations.** The augmentation operations are summarized in Table 2. Please refer to Appendix C.5 for detailed descriptions. We first conduct experiments to compare different contrastive operations on hypergraphs, with results shown in Table 3. In general, generalized hyperedge augmentation (A2) works the best among fabricated augmenting operators, but not naïvely perturbing hyperedge (A1). Specifically, among all fabricated augmentations, A2 performs the best in 10 of 13 data sets. This indicates the nature that higher-order information in structures is usually more downstream-relevant.

Table 2: The proposed augmentation operations for contrastive learning framework HyperGCL and their corresponding names.

| Name | Operation |
|---|---|
| A0 | Identity |
| A1 | Naïve Hyperedge Perturbation |
| A2 | Generalized Hyperedge Perturbation |
| A3 | Vertex Dropping |
| A4 | Attribute Masking |
| A5 | Subgraph |
| A6 | Generative Augmentation |

For our generative augmentation (A6), we find it performs the best in all the data sets. In our joint augmenting and contrasting framework, we generate stronger augmentation while keeping the hypergraph distribution with adversarial learning. This illustrates the importance of exploring the hypergraph structure. We also test our method on 1% label setting in Table 4. In this setting, Zoo and NTU2012 data sets are not shown because of the extremely small data size (each case has less than one training sample). We can find that in the 1% label setting A4 (mask) method performs the best in Cora, Citeseer and DBLP-CA. These data sets are all originally graphs, and are constructed as hypergraphs in different ways. So on these data sets, relatively little higher-order structural information can be explored with hypergraph structure perturbation-based contrastive learning.

**Comparison between multi-task learning and pretraining.** We then compare the multi-task training method with the pretraining method in Table 5. Pretrain_L adopts the linear evaluation protocol where a linear classifier is trained on top of the fixed pretrained representations. Pretrain_F follows a fully finetuning protocol that uses the weights of the learned hypergnn encoder as initialization while finetuning all the layers. MTL denotes the multi-task learning method which trains the supervised classification loss and contrastive loss together. For all the methods, we use A2 (generalized hyperedge perturbation) as it performs the best among fabricated augmentations. From the table, we find MTL achieves the best performance in nearly all data sets. Pretrain_L and Pretrain_F can only obtain better performance on two small data sets: Zoo and House. We find on most data sets, Pretrain_L makes the model perform worse, which shows that the linear classifier is not enough to represent the higher-order information in the hypergraph. Pretrain_F has a much better performance compared

Table 3: Results on the test data sets: Mean accuracy (%) ± standard deviation. Bold values indicate the best result. Underlined values indicate the second best. 10% of all vertices are used for training.

| | Cora | Citeseer | Pubmed | Cora-CA | DBLP-CA | Zoo | 20Newsgroups | Mushroom |
|---|---|---|---|---|---|---|---|---|
| SetGNN | 67.93 ± 1.27 | 63.53 ± 1.32 | 84.33 ± 0.36 | 72.21 ± 1.51 | 89.51 ± 0.18 | 65.06 ± 12.82 | 79.37 ± 0.35 | 99.75 ± 0.11 |
| Self | 68.24 ± 1.12 | 62.49 ± 1.48 | 84.38 ± 0.38 | 72.74 ± 1.53 | 89.51 ± 0.23 | 57.35 ± 18.32 | 79.45 ± 0.32 | 95.83 ± 0.23 |
| Con | 68.89 ± 1.80 | 62.82 ± 1.21 | 84.56 ± 0.34 | 73.22 ± 1.65 | 89.59 ± 0.13 | 61.05 ± 14.54 | 79.49 ± 0.45 | 95.85 ± 0.31 |
| A0 | 68.59 ± 1.33 | 62.25 ± 2.15 | 84.54 ± 0.42 | 71.85 ± 1.62 | 89.62 ± 0.24 | 62.57 ± 13.84 | 79.07 ± 0.46 | 99.77 ± 0.17 |
| A1 | 72.39 ± 1.34 | 66.28 ± 1.27 | 85.17 ± 0.37 | 75.45 ± 1.54 | 89.83 ± 0.21 | 65.80 ± 13.31 | 79.47 ± 0.32 | 99.80 ± 0.14 |
| A2 | 72.58 ± 1.09 | 66.40 ± 1.35 | 85.16 ± 0.38 | 75.62 ± 1.42 | 90.22 ± 0.23 | 66.35 ± 13.26 | 79.56 ± 0.42 | 99.80 ± 0.17 |
| A3 | 72.33 ± 1.23 | 65.79 ± 1.18 | 85.24 ± 0.28 | 75.34 ± 1.40 | 89.85 ± 0.16 | 65.79 ± 14.05 | 79.47 ± 0.34 | 99.81 ± 0.10 |
| A4 | 72.95 ± 1.19 | 66.22 ± 0.95 | 84.88 ± 0.38 | 75.29 ± 1.56 | 90.10 ± 0.18 | 62.59 ± 12.77 | 79.45 ± 0.48 | 99.80 ± 0.14 |
| A5 | 67.96 ± 0.99 | 63.21 ± 1.25 | 84.48 ± 0.40 | 72.61 ± 1.86 | 89.75 ± 0.24 | 62.47 ± 12.39 | 79.42 ± 0.52 | 99.79 ± 0.10 |
| A6 | **73.12 ± 1.48** | **66.94 ± 1.00** | **85.72 ± 0.38** | **76.21 ± 1.26** | **90.28 ± 0.19** | **66.89 ± 12.44** | **79.78 ± 0.40** | **99.86 ± 0.10** |

| | NTU2012 | ModelNet40 | Yelp | House (0.6) | House (1.0) | Walmart (0.6) | Walmart (1.0) | Avg. Rank |
|---|---|---|---|---|---|---|---|---|
| SetGNN | 73.86 ± 1.62 | 95.85 ± 0.38 | 28.78 ± 1.51 | 68.54 ± 1.89 | 58.34 ± 2.25 | 74.97 ± 0.22 | 59.13 ± 0.20 | 7.71 |
| Self | 73.41 ± 1.65 | 95.83 ± 0.23 | 23.49 ± 4.15 | 67.75 ± 3.29 | 58.54 ± 2.16 | 74.76 ± 0.20 | 58.83 ± 0.21 | 8.64 |
| Con | 73.27 ± 1.53 | 95.85 ± 0.31 | 26.14 ± 1.86 | 68.50 ± 2.52 | 58.56 ± 2.42 | 75.17 ± 0.21 | 59.39 ± 0.20 | 7.07 |
| A0 | 73.54 ± 1.93 | 95.92 ± 0.18 | 29.43 ± 1.42 | 67.48 ± 3.21 | 57.39 ± 2.37 | 73.14 ± 0.21 | 56.49 ± 0.60 | 8.21 |
| A1 | 74.71 ± 1.81 | 95.87 ± 0.27 | 27.18 ± 0.71 | 68.64 ± 2.99 | 58.10 ± 3.22 | 75.42 ± 0.13 | 60.09 ± 0.25 | 4.50 |
| A2 | 74.88 ± 1.66 | 96.56 ± 0.34 | 31.39 ± 2.45 | 69.73 ± 2.60 | 58.90 ± 1.97 | 75.50 ± 0.18 | 60.19 ± 0.20 | 2.29 |
| A3 | 74.68 ± 1.74 | 96.48 ± 0.29 | 27.57 ± 1.00 | 67.88 ± 2.90 | 58.51 ± 2.22 | 75.29 ± 0.23 | 60.19 ± 0.20 | 4.71 |
| A4 | 74.83 ± 1.75 | 95.86 ± 0.28 | 29.64 ± 1.93 | 69.56 ± 2.89 | 58.91 ± 2.69 | 75.43 ± 0.18 | 59.90 ± 0.24 | 4.14 |
| A5 | 74.41 ± 1.86 | 96.46 ± 0.33 | 29.24 ± 1.42 | 68.14 ± 2.97 | 57.70 ± 2.98 | 75.26 ± 0.18 | 59.81 ± 0.22 | 6.71 |
| A6 | **75.34 ± 1.91** | **96.93 ± 0.33** | **34.64 ± 0.39** | **70.96 ± 2.27** | **59.93 ± 1.99** | **75.62 ± 0.16** | **60.46 ± 0.20** | **1.00** |

Table 4: Results on the test data sets: Mean accuracy (%) ± standard deviation. Bold values indicate the best result. 1% of all vertexes are used for training.

| | Cora | Citeseer | Pubmed | Cora-CA | DBLP-CA | 20Newsgroups | Mushroom |
|---|---|---|---|---|---|---|---|
| SetGNN | 46.48 ± 3.62 | 47.01 ± 4.31 | 76.13 ± 1.19 | 52.29 ± 4.18 | 85.52 ± 0.54 | 73.83 ± 1.40 | 97.73 ± 1.18 |
| Self | 45.79 ± 5.34 | 44.22 ± 4.43 | 76.71 ± 0.90 | 51.64 ± 5.37 | 84.42 ± 0.37 | 73.91 ± 0.90 | 92.25 ± 0.89 |
| Con | 49.20 ± 4.38 | 48.56 ± 4.88 | 77.51 ± 1.08 | 52.37 ± 4.41 | 86.47 ± 0.35 | 74.39 ± 1.23 | 92.43 ± 0.87 |
| A0 | 48.50 ± 4.77 | 46.43 ± 4.24 | 78.83 ± 1.79 | 49.87 ± 5.08 | 87.34 ± 0.73 | 74.43 ± 1.11 | 97.32 ± 1.33 |
| A1 | 56.42 ± 5.02 | 55.63 ± 3.96 | 80.13 ± 1.44 | 60.86 ± 5.91 | 87.53 ± 0.30 | 74.68 ± 1.31 | 97.95 ± 1.15 |
| A2 | 56.81 ± 4.49 | 56.10 ± 2.86 | 80.22 ± 1.24 | 60.96 ± 6.31 | 88.10 ± 0.35 | 74.72 ± 1.16 | 98.05 ± 1.18 |
| A3 | 55.94 ± 3.67 | 55.82 ± 3.40 | 80.13 ± 1.02 | 60.51 ± 4.55 | 87.47 ± 0.36 | 74.63 ± 1.00 | 98.04 ± 0.98 |
| A4 | 58.55 ± 5.14 | 57.16 ± 4.62 | 80.11 ± 1.02 | 60.91 ± 5.15 | 88.91 ± 0.29 | 74.67 ± 1.39 | 97.72 ± 1.12 |
| A5 | 46.23 ± 3.44 | 45.07 ± 4.89 | 75.95 ± 1.32 | 53.26 ± 4.86 | 87.12 ± 0.43 | 74.81 ± 1.04 | 97.72 ± 1.25 |
| A6 | 57.45 ± 5.00 | 56.23 ± 3.27 | **81.10 ± 0.80** | **61.76 ± 4.94** | 88.55 ± 0.41 | **75.52 ± 0.93** | **98.28 ± 1.03** |

| | ModelNet40 | Yelp | House (0.6) | House (1.0) | Walmart (0.6) | Walmart (1.0) | Avg. Rank (↓) |
|---|---|---|---|---|---|---|---|
| SetGNN | 88.34 ± 2.69 | 27.64 ± 1.10 | 53.69 ± 2.20 | 51.85 ± 1.64 | 65.48 ± 0.45 | 51.15 ± 0.52 | 7.62 |
| Self | 86.85 ± 3.03 | 20.77 ± 5.15 | 53.42 ± 2.25 | 51.14 ± 1.75 | 65.23 ± 0.43 | 51.00 ± 0.41 | 9.69 |
| Con | 87.00 ± 2.99 | 24.23 ± 0.43 | 53.58 ± 3.04 | 51.96 ± 1.87 | 65.47 ± 0.44 | 51.13 ± 0.46 | 7.31 |
| A0 | 88.75 ± 2.78 | 27.43 ± 0.60 | 53.60 ± 2.73 | 51.70 ± 2.13 | 65.41 ± 0.47 | 51.10 ± 0.49 | 7.46 |
| A1 | 89.34 ± 2.66 | 26.18 ± 0.51 | 54.12 ± 3.29 | 52.23 ± 2.46 | 65.96 ± 0.36 | 51.22 ± 0.35 | 4.08 |
| A2 | 89.37 ± 2.69 | 27.67 ± 0.91 | 54.42 ± 2.83 | 52.31 ± 1.44 | 66.01 ± 0.41 | 51.32 ± 0.30 | 2.69 |
| A3 | 89.31 ± 2.62 | 26.98 ± 0.66 | 53.71 ± 2.71 | 52.11 ± 2.24 | 65.88 ± 0.50 | 51.35 ± 0.53 | 4.38 |
| A4 | 89.03 ± 2.66 | 27.45 ± 0.81 | 53.64 ± 2.61 | 51.77 ± 2.20 | 65.55 ± 0.51 | 51.04 ± 0.47 | 4.54 |
| A5 | 89.43 ± 2.68 | 28.09 ± 0.96 | 54.07 ± 3.09 | 51.94 ± 1.84 | 65.52 ± 0.39 | 50.97 ± 0.47 | 6.00 |
| A6 | **90.22 ± 2.72** | **29.61 ± 0.71** | **56.27 ± 4.18** | **52.55 ± 2.18** | **66.42 ± 0.40** | **51.82 ± 0.39** | **1.23** |

with Pretrain_L, which indicates the effects of using contrastive learning. However, the method switches the objective during finetuning, which would lead to the memorization problem and the loss of pre-trained knowledge. Therefore, all our other experiments adopt MTL setting to incorporate contrastive self-supervision.

**Comparison to the converted graph.** Next we investigate on which graph should we contrast on. We first convert the original hypergraph into a conventional graph using the clique expansion technique, and we choose the representative HGNN[1] as the backbone network for learning on the converted graph. We compare it with two representative augmentations: A2 (Here, edge perturbation on the conventional graph) and A4 (feature masking). In Table 6, we find that HGNN performs poorly compared with SetGNN. Apparently contrastive self-supervision on converted graphs does not bring much benefit. The reason is that part of structural information, important to hypergraph representation learning, is lost when hypergraphs are converted to graphs. These results indicate the importance of designing HyperGNN and contrastive strategies directly on hypergraphs.

Table 5: Results of different self-supervised mechanisms: Mean accuracy (%) ± standard deviation. Bold values indicate the best result. 10% of all vertexes are used for training.

| | Cora | Citeseer | Pubmed | Cora-CA | DBLP-CA | Zoo | 20Newsgroups |
|---|---|---|---|---|---|---|---|
| SetGNN | 67.93 ± 1.27 | 63.53 ± 1.32 | 84.33 ± 0.36 | 72.21 ± 1.51 | 89.51 ± 0.18 | 65.06 ± 12.82 | 79.37 ± 0.35 |
| Pretrain_L | 52.59 ± 2.33 | 53.29 ± 2.01 | 69.90 ± 0.41 | 48.00 ± 4.79 | 87.59 ± 0.43 | **66.82 ± 13.48** | 71.93 ± 2.99 |
| Pretrain_F | 68.39 ± 1.20 | 63.83 ± 1.68 | 84.47 ± 0.40 | 73.12 ± 1.37 | 89.75 ± 0.23 | 65.43 ± 13.38 | 79.44 ± 0.39 |
| MTL | **72.58 ± 1.10** | **66.40 ± 1.35** | **85.16 ± 0.38** | **75.82 ± 1.42** | **90.22 ± 0.23** | 66.35 ± 13.26 | **79.56 ± 0.42** |

| | Mushroom | NTU2012 | ModelNet40 | Yelp | House (0.6) | House (1.0) | Walmart (0.6) | Walmart (1.0) |
|---|---|---|---|---|---|---|---|---|
| | 99.75 ± 0.11 | 73.86 ± 1.62 | 95.85 ± 0.38 | 28.78 ± 1.51 | 68.54 ± 1.89 | 58.34 ± 2.25 | 74.97 ± 0.22 | 59.13 ± 0.20 |
| | 93.77 ± 2.20 | 70.06 ± 2.42 | 96.23 ± 0.31 | 26.68 ± 0.30 | 61.22 ± 3.09 | 54.81 ± 2.39 | 40.35 ± 4.30 | 33.30 ± 2.72 |
| | 99.77 ± 0.15 | 74.03 ± 1.86 | 95.88 ± 0.34 | 28.19 ± 1.42 | 69.02 ± 4.02 | 59.20 ± 2.54 | 75.01 ± 0.27 | 59.87 ± 0.28 |
| | **99.80 ± 0.17** | **74.88 ± 1.66** | **96.56 ± 0.34** | **31.39 ± 2.45** | **69.73 ± 2.60** | 58.90 ± 1.97 | **75.50 ± 0.18** | **60.19 ± 0.20** |

Table 6: Results on converted conventional graphs: Mean accuracy (%) ± standard deviation. Bold values indicate the best result. 10% of all vertices are used for training.

| | Cora | Citeseer | Pubmed | Cora-CA | DBLP-CA | Zoo | 20Newsgroups |
|---|---|---|---|---|---|---|---|
| HGNN | **67.37 ± 1.45** | 62.76 ± 1.42 | 82.16 ± 0.38 | 66.80 ± 1.79 | **85.28 ± 0.29** | **47.84 ± 6.87** | 70.27 ± 0.73 |
| A2 | 67.18 ± 1.42 | **63.52 ± 2.35** | **82.37 ± 0.34** | **67.14 ± 1.79** | 85.22 ± 0.26 | 46.85 ± 10.15 | **70.46 ± 1.21** |
| A4 | 67.24 ± 1.51 | 63.37 ± 2.56 | 82.25 ± 0.43 | 66.88 ± 2.07 | 85.16 ± 0.25 | 46.85 ± 9.93 | 69.35 ± 1.24 |

| | Mushroom | NTU2012 | ModelNet40 | Yelp | House (0.6) | House (1.0) | Walmart (0.6) | Walmart (1.0) |
|---|---|---|---|---|---|---|---|---|
| | 97.15 ± 0.47 | **70.26 ± 1.70** | 87.60 ± 0.36 | **26.91 ± 0.37** | 58.01 ± 2.47 | 57.65 ± 2.69 | 59.48 ± 0.19 | 53.97 ± 0.29 |
| | **97.29 ± 0.45** | 69.91 ± 1.59 | **87.75 ± 0.33** | 26.72 ± 0.36 | 58.08 ± 3.28 | 57.53 ± 2.80 | 59.49 ± 0.22 | **54.04 ± 0.24** |
| | 97.15 ± 0.55 | 69.94 ± 1.54 | 87.65 ± 0.36 | 26.66 ± 0.45 | **58.47 ± 2.99** | **57.73 ± 2.84** | **59.53 ± 0.22** | 53.98 ± 0.26 |

**Analysis of generative augmentation.** Here we analyze our proposed generative hypergraph augmentation (A6). We select ModelNet40 and Yelp as the representatives of high-homophily and low-homophily data sets, respectively.

First, we examine the training dynamics of keep ratio and find that they are highly related to the data set homophily (Figure 4 (a)). For ModelNet40, the generator keeps only 20% relations in the early training stage. This is because the homophily of the data set is very high (0.92/0.88) and deleting a lot of edges relatively randomly at the beginning would not have a large impact on the model but rather can help model learn structural information. Then at a later stage, the generator begins to keep more than 80% relations, which means that the model has learned higher-order information and only removes the unnecessary relations. For Yelp, a similar conclusion holds. Specifically, as its homophily is pretty low (0.57/0.26), the generator keeps most of the relations at the early stage for training and then just keeps a very low ratio of related relations at the later training stage.

Next, we investigate what our generator has learned. We visualize the hyperedges in the Yelp data set in Figure 4 (b). Yelp is a restaurant-rating data set and the restaurants visited by the same user are connected by a hyperedge. We find that some vertices with different labels are removed, which could remove extraneous information and improve the hypergraph homophily. This indicates that our generator does grasp the higher-order information in the hypergraph.

**Adversarial robustness.** Besides generalizability, we here show hypergraph contrastive learning also boosts robustness. Since there is no existing work developing adversarial attack algorithms designated for hypergraphs, we adapt two state-of-the-art attackers from the graph domain. We presume the graph attackers are applicable to hypergraphs to a certain extent, both of which are non-Euclidean data structures with sharing properties. The experiments are performed on five real-world data sets: Cora, Citeseer, ModelNet40, NTU2012 and House. We regard each hypergraph as a bipartite graph and leverage those algorithms to conduct attacks to the vertices and hyperedges. The methods include an untargeted attack method, minmax attack [46] which poisons the graph structures by adding and removing relations to reduce the overall performance; and a targeted attack method, nettack [47] which leads the HyperGNN to mis-classify target vertices. Beyond these, we also include a random hypergraph perturbation baseline which will randomly drop numerous relations in the hypergraph. These three methods are denoted as Net, Minmax and Random. Following previous works, for each attack, we perturb 10% of vertices/relations. The results in Table 7 show that Random and Minmax attacks can decrease the performance of the original model a little on all the data sets, while net attack can decrease the performance on most data sets and surprisingly increase the performance on

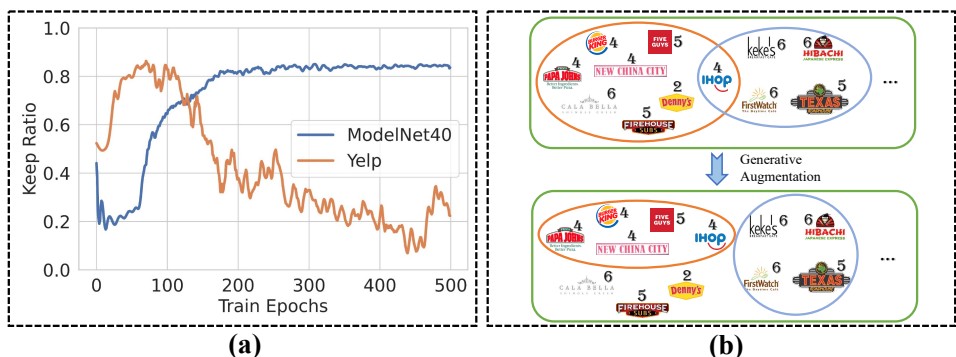

**(a)**                    **(b)**

Figure 4: (a) Training dynamics of the relation keep ratio. (b) Illustration of our proposed generative augmentation on the Yelp data set. Each icon represents a restaurant in the data set, and the number near the icon is the label of this restaurant. The ellipse denotes the hyperedge.

Table 7: Results on the test data sets with regard to robustness. Bold values indicate the best result. 10% of all vertexes are used for training.

| | Cora | | | Citeseer | | | ModelNet40 | | |
|---|---|---|---|---|---|---|---|---|---|
| | Random | Net | Minmax | Random | Net | Minmax | Random | Net | Minmax |
| SetGNN | 66.87 ± 1.33 | 66.26 ± 1.54 | 66.58 ± 1.02 | 62.89 ± 1.57 | 62.81 ± 1.32 | 62.21 ± 1.64 | 95.74 ± 0.22 | 95.41 ± 0.28 | 93.33 ± 0.26 |
| A2 | 71.90 ± 1.63 | 71.16 ± 0.92 | 70.86 ± 1.22 | 66.41 ± 1.08 | 65.38 ± 1.47 | 64.69 ± 0.98 | 96.09 ± 0.17 | 95.52 ± 0.24 | 93.64 ± 0.26 |
| A4 | 72.11 ± 1.60 | 70.49 ± 1.29 | 70.52 ± 1.39 | 65.94 ± 1.24 | 65.15 ± 1.70 | 64.12 ± 1.19 | 95.79 ± 0.27 | 95.44 ± 0.25 | 93.35 ± 0.24 |
| A6 | **72.15 ± 1.70** | **71.94 ± 1.48** | **71.98 ± 1.36** | **66.60 ± 1.61** | **65.68 ± 1.09** | **65.51 ± 1.13** | **96.58 ± 0.24** | **96.23 ± 0.23** | **94.82 ± 0.33** |
| | NTU2012 | | | House (0.6) | | | House (1.0) | | |
| | Random | Net | Minmax | Random | Net | Minmax | Random | Net | Minmax |
| SetGNN | 73.84 ± 2.18 | 73.38 ± 1.36 | 70.71 ± 1.89 | 67.16 ± 2.55 | 68.88 ± 2.68 | 64.78 ± 2.20 | 56.86 ± 1.93 | 59.95 ± 1.92 | 56.52 ± 2.52 |
| A2 | 74.50 ± 2.03 | 73.86 ± 1.84 | 71.40 ± 1.64 | 67.71 ± 2.94 | 69.59 ± 2.32 | 65.23 ± 2.89 | 57.74 ± 2.70 | 60.73 ± 2.30 | 57.00 ± 1.94 |
| A4 | 73.73 ± 1.59 | 73.72 ± 1.59 | 71.06 ± 1.53 | 67.55 ± 2.41 | 68.85 ± 1.38 | 64.97 ± 3.35 | 57.47 ± 2.72 | 60.10 ± 1.74 | 56.65 ± 2.26 |
| A6 | **75.06 ± 1.97** | **74.37 ± 1.99** | **72.09 ± 1.98** | **69.88 ± 3.27** | **73.14 ± 2.71** | **68.84 ± 2.71** | **60.06 ± 2.07** | **62.41 ± 1.77** | **58.76 ± 2.24** |

the House data set. This performance gain indicates HyperGNN is more robust to structure attack compared with GNN as it leverages higher-order information. Based on these, HyperGCL with generalized hyperedge augmentation (A2) performs better than feature perturbation (A4), and our proposed generative augmentation (A6) can surpass these fabricated baselines on all the data sets and thus is the best to defend attacks. We believe this'll be a beneficial complement to our main experiments and we hope for more works on hypergraph attacks.

**Fairness.** Furthermore, we claim that hypergraph contrastive self-supervision also benefits fairness. There was no related data set before. So we introduce three newly curated hypergraph data sets: German [48], Recidivism [49] and Credit [50]. The hypergraph construction follows the setting in [1]. The top 5 similar objects in each data set are built as a hyperedge. For the accuracy metrics, we use F1-score and AUROC value for the binary classification task. For measuring fairness, we adopt the statistical parity $\Delta_{SP}$ and equalized odds $\Delta_{EO}$. Please refer to Appendix C for detailed information about the data sets and metrics. The experimental results in Table 8 show that our generative method still achieves better or comparable performances while imposing more fairness.

## 5 Conclusion

In the paper, we study the problem of how to construct contrastive views of hypergraphs via augmentations. We provide the solutions by first studying domain knowledge-guided fabrication schemes. Then, in search of more effective views in a data-driven manner, we are the first to propose hypergraph generative models to generate augmented views, as well as an end-to-end differentiable pipeline to jointly perform hypergraph augmentation and contrastive learning. We find that generative augmentations perform better at preserving higher-order information to further benefit generalizability. The proposed framework also boosts robustness and fairness of hypergraph representation learning. In the future, we plan to design more powerful hypergraph generator and HyperGNN while addressing more real-world hypergraph data challenges and more hypergraph learning models.

Table 8: Results on the test data sets with regard to fairness. 10% of all vertexes are used for training. For fairness metrics $\Delta_{SP}$ and $\Delta_{EO}$, lower values indicate better performance.

| data set | Method | AUROC | F1 | $\Delta_{SP}(\downarrow)$ | $\Delta_{EO}(\downarrow)$ |
|---|---|---|---|---|---|
| German Credit | SetGNN | $59.16 \pm 2.51$ | $81.84 \pm 0.93$ | $2.65 \pm 5.62$ | $4.06 \pm 6.76$ |
| | A2 | $\underline{59.81 \pm 3.00}$ | $\underline{82.26 \pm 0.13}$ | $\mathbf{0.55 \pm 0.95}$ | $\underline{0.78 \pm 0.70}$ |
| | A4 | $59.66 \pm 3.83$ | $80.54 \pm 3.52$ | $3.03 \pm 6.54$ | $5.07 \pm 7.81$ |
| | A6 | $\mathbf{59.88 \pm 3.04}$ | $\mathbf{82.36 \pm 0.38}$ | $\underline{0.95 \pm 0.92}$ | $\mathbf{0.47 \pm 0.56}$ |
| Recidivism | SetGNN | $\underline{96.51 \pm 0.48}$ | $\underline{89.84 \pm 0.97}$ | $8.63 \pm 0.50$ | $4.16 \pm 0.51$ |
| | A2 | $96.34 \pm 0.39$ | $\mathbf{90.09 \pm 0.53}$ | $8.53 \pm 0.52$ | $3.92 \pm 0.68$ |
| | A4 | $96.45 \pm 0.35$ | $89.75 \pm 0.68$ | $\mathbf{8.49 \pm 0.27}$ | $\underline{3.49 \pm 0.66}$ |
| | A6 | $\mathbf{96.55 \pm 0.54}$ | $89.22 \pm 0.55$ | $\underline{8.51 \pm 0.25}$ | $\mathbf{3.13 \pm 0.64}$ |
| Credit defaulter | SetGNN | $73.46 \pm 0.17$ | $87.91 \pm 0.27$ | $2.79 \pm 0.99$ | $0.98 \pm 0.69$ |
| | A2 | $73.43 \pm 0.27$ | $87.82 \pm 0.24$ | $\underline{2.64 \pm 1.32}$ | $\underline{0.93 \pm 0.87}$ |
| | A4 | $\underline{73.58 \pm 0.19}$ | $\underline{87.92 \pm 0.25}$ | $2.84 \pm 1.14$ | $1.38 \pm 0.32$ |
| | A6 | $\mathbf{73.78 \pm 0.16}$ | $\mathbf{88.03 \pm 0.14}$ | $\mathbf{2.58 \pm 0.91}$ | $\mathbf{0.81 \pm 0.37}$ |

## Acknowledgments and Disclosure of Funding

This work is supported by National Science Foundation under Award No. IIS-1947203, IIS-2117902, IIS-2137468, CCF-1943008; US Army Research Office Young Investigator Award W911NF2010240; National Institute of General Medical Sciences under grant R35GM124952; and Agriculture and Food Research Initiative grant no. 2020-67021-32799/project accession no.1024178 from the USDA National Institute of Food and Agriculture. The views and conclusions are those of the authors and should not be interpreted as representing the official policies of the government agencies.

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
