# Appendix:
# Augmentations in Hypergraph Contrastive Learning: Fabricated and Generative

**Tianxin Wei**[1*], **Yuning You**[2*], **Tianlong Chen**[3], **Yang Shen**[2], **Jingrui He**[1], **Zhangyang Wang**[3]

[1]University of Illinois Urbana-Champaign, [2]Texas A&M University, [3]University of Texas at Austin

{twei10, jingrui}@illinois.edu, {yuning.you,yshen}@tamu.edu,
{tianlong.chen,atlaswang}@utexas.edu

## A  Hyperparameter Study

In this section, we conduct experiments to explore the effect of hyperparameters. There are two important tradeoff parameters $\alpha$, and $\beta$ in our proposed method. We select four representative datasets to perform the ablation study. For each data set, when varying one parameter, the other is set as constant.

Table 1: Results of different $\alpha$.

| $\alpha$ | Pubmed | Cora-CA | Yelp | House |
|---|---|---|---|---|
| 0.1 | 85.62 ± 0.43 | 75.93 ± 1.39 | 33.92 ± 0.58 | 59.54 ± 2.07 |
| 0.2 | 85.63 ± 0.37 | 76.02 ± 1.65 | **34.64 ± 0.39** | 59.65 ± 2.25 |
| 0.5 | **85.72 ± 0.38** | **76.21 ± 1.26** | 33.71 ± 1.40 | **59.93 ± 1.99** |
| 1.0 | 84.97 ± 0.37 | 75.12 ± 1.46 | 31.54 ± 1.29 | 57.70 ± 2.38 |

For $\alpha$, it is a tradeoff factor used in multi-task learning loss $\mathcal{L}_{sup} + \alpha \cdot \mathcal{L}_{cl}(\tilde{\mathcal{G}}_p, \tilde{\mathcal{G}}_{gen})$ to balance the supervised classification loss and contrastive learning loss. To investigate the effect of $\alpha$, we search its value in the range of {0.1, 0.2, 0.5, 1.0}. The experimental results are summarized in Table 1. From the table, we can find that $\alpha$ is able to improve the performance in a wide range of hyper-parameters (0.1-0.5). However, when it surpasses a threshold, the performance becomes worse with a further increase of the parameter. As the parameter becomes even larger, the optimization of the classification loss will be less important, which brings worse results.

Table 2: Results of different $\beta$.

| $\beta$ | Pubmed | Cora-CA | Yelp | House |
|---|---|---|---|---|
| 0 | 85.23 ± 0.45 | 75.71 ± 1.57 | 31.82 ± 0.55 | 58.91 ± 1.93 |
| 1 | **85.72 ± 0.38** | 76.07 ± 1.38 | 32.91 ± 0.44 | 58.96 ± 2.25 |
| 2 | 85.61 ± 0.41 | **76.21 ± 1.26** | 33.57 ± 0.42 | 59.27 ± 3.04 |
| 5 | 85.34 ± 0.42 | 75.49 ± 1.39 | 34.05 ± 0.35 | 59.47 ± 1.85 |
| 10 | 85.25 ± 0.35 | 75.02 ± 1.26 | **34.64 ± 0.39** | **59.93 ± 1.99** |

For $\beta$, it is the tradeoff factor in the loss designed to optimize the generator. The goal of the loss $\mathcal{L}_{gen} - \beta \cdot \mathcal{L}_{cl}(\mathcal{G}, \tilde{\mathcal{G}}_{gen})$ is to generate stronger augmentation (maximizing contrastive loss) for pushing HyperGNN to avoid capturing redundant information during the representation learning, while at the same time learning the hypergraph data distribution ($\mathcal{L}_{gen}$). The experimental results are listed

---

*Equal contribution.

36th Conference on Neural Information Processing Systems (NeurIPS 2022).

in Table 2. In the table, we can see that setting $\beta$ to 0 only improves the performance a little. This illustrates the importance of removing redundant information when designing the generator. Then we find the value of $\beta$ is related to the homophily of the data set. For high-homophily data sets Pubmed and Cora-CA, setting a smaller value of $\beta$ will lead to better performance as the hypergraph is more homogeneous and there are fewer unrelated relations. We can observe a similar phenomenon on the low-homophily data sets Yelp and House, i.e. setting a larger $\beta$ value will work better.

## B  Model Details

### B.1  Contrastive Loss

Given a hypergraph $\mathcal{G}$, two contrastive views $\tilde{\mathcal{G}}_1$ and $\tilde{\mathcal{G}}_2$ are first generated by hypergraph data augmentations $\mathcal{A}$. The vertex projected embedding for each hypergraph view can be obtained as $Z_1^{\mathcal{V}} = h \circ f(\tilde{\mathcal{G}}_1)$, $Z_2^{\mathcal{V}} = h \circ f(\tilde{\mathcal{G}}_2)$. In the two hypergraph views, the corresponding vertex pairs are positive pairs, while all other vertex pairs are denoted as negative. The $n$-th vertex embedding in the two views is represented as $\mathbf{u}_n = Z_1^{\mathcal{V}}[n,:]$ and $\mathbf{s}_n = Z_2^{\mathcal{V}}[n,:]$. Given the cosine similarity function $\gamma(\mathbf{u},\mathbf{s}) = \frac{\mathbf{u}^T\mathbf{s}}{||\mathbf{u}||||\mathbf{s}||}$, the contrastive loss can be constructed as:

$$\mathcal{L}_{cl}(\tilde{\mathcal{G}}_1, \tilde{\mathcal{G}}_2) = \frac{1}{2|\mathcal{V}|} \sum_{n=1}^{|\mathcal{V}|} (l(\mathbf{u}_n, \mathbf{s}_n) + l(\mathbf{s}_n, \mathbf{u}_n)), \tag{1}$$

$$l(\mathbf{u}_n, \mathbf{s}_n) = -\log \frac{e^{\gamma(\mathbf{u}_n, \mathbf{s}_n)/\tau}}{e^{\gamma(\mathbf{u}_n, \mathbf{s}_n)/\tau} + \sum_{m \neq n} e^{\gamma(\mathbf{u}_n, \mathbf{s}_m)/\tau} + \sum_{m \neq n} e^{\gamma(\mathbf{u}_n, \mathbf{u}_m)/\tau}}, \tag{2}$$

where $\tau$ is a temperature parameter. The loss $l$ is symmetrically defined. The contrastive loss could be applied on any HyperGNN architectures.

### B.2  Training Pipeline

---
**Algorithm 1** Hypergraph Contrastive Learning with Generative Augmentation (A6)

---
**Input:** Hypergraph $\mathcal{G}$; HyperGNN and generator parameters $\theta$ and $\phi$; Multi-task training tradeoff parameters $\alpha, \beta$

1: Randomly initialize $\theta$ and $\phi$;
2: **while** not converge **do**
3:   Obtain view $\tilde{\mathcal{G}}_p$ via fabricated augmentation and view $\tilde{\mathcal{G}}_{gen}$ via generator $\phi$;
4:   Define HyperGNN loss as: $\mathcal{L}_h = \mathcal{L}_{sup}(\theta) + \alpha \cdot \mathcal{L}_{cl}(\tilde{\mathcal{G}}_p, \tilde{\mathcal{G}}_{gen} \mid \theta, \phi)$;
5:   Define generator loss as: $\mathcal{L}_g = \mathcal{L}_{gen}(\phi) - \beta \cdot \mathcal{L}_{cl}(\mathcal{G}, \tilde{\mathcal{G}}_{gen} \mid \theta, \phi)$;
6:   Update HyperGNN $\theta$ to minimize $\mathcal{L}_h$;
7:   Update generator $\phi$ to minimize $\mathcal{L}_g$;
8: **end while**

---

In this section, we describe the training process of our proposed generative augmentation (A6) in detail. The HyperGNN model and generator model parameters are denoted as $\theta$ and $\phi$ respectively. First, the generator will sample the relations to produce the generated hypergraph $\tilde{\mathcal{G}}_{gen}$. To optimize HyperGNN, the multi-task training loss is formulated as:

$$\min_\theta \mathcal{L}_{sup}(\theta) + \alpha \cdot \mathcal{L}_{cl}(\tilde{\mathcal{G}}_p, \tilde{\mathcal{G}}_{gen} \mid \theta, \phi), \tag{3}$$

where $\mathcal{L}_{sup}$ is the supervised classification loss, $\mathcal{L}_{cl}$ is the contrastive learning loss, $\alpha$ is the tradeoff factor to balance these two losses. Due to the expensive computational cost of VHGAE, we train one VHGAE to produce one generative view $\tilde{\mathcal{G}}_{gen}$, while the other view $\tilde{\mathcal{G}}_p$ is kept as fabricated.

Then the generator will be optimized as:

$$\min_\phi \mathcal{L}_{gen}(\phi) - \beta \cdot \mathcal{L}_{cl}(\mathcal{G}, \tilde{\mathcal{G}}_{gen} \mid \theta, \phi), \tag{4}$$

where $\mathcal{L}_{gen}$ is the evidence lower bound of the variational generator, $\mathcal{L}_{cl}$ is the contrastive learning loss, and $\beta$ is the tradeoff factor to balance these two losses. Here we maximize the contrastive loss to encourage the generative view to share less mutual information. It can push HyperGNN to avoid capturing redundant information during the representation learning to facilitate downstream generalization. In this way, the learned information will be more robust and transferable. Our ablation study in Section A has demonstrated its effectiveness. The full pipeline is shown in Algorithm 1.

### B.3 Details of the Proposed VHGAE

**Encoder** The goal of the generator is to learn the hypergraph distribution $p_\theta(\mathcal{G})$. However, the inference of the true distribution is intractable, thus we introduce the latent variables $z_\mathcal{E}$, $z_\mathcal{V}$ and their corresponding variational posterior distributions $q_\phi(z_\mathcal{E}|\mathcal{G})$, $q_\phi(z_\mathcal{V}|\mathcal{G})$, where $z_\mathcal{E}$, $z_\mathcal{V}$ can be regarded as the hidden representations of hyperedges and vertexes. For latent variables $z_\mathcal{E}$ and $z_\mathcal{V}$, we consider the multivariate normal variational posterior $z_\mathcal{E} \sim \mathcal{N}\left(z^\mathcal{E}|\mu_\mathcal{E}, \sigma_\mathcal{E}^2 I\right)$ and $z_\mathcal{V} \sim \mathcal{N}\left(z_\mathcal{V}|\mu_\mathcal{V}, \sigma_\mathcal{V}^2 I\right)$, where where $\mu$, $\sigma^2$ are the mean and covariance of the Gaussian distribution. We leverage the hypergraph neural network to parameterize the above posterior and infer $z_\mathcal{E}$, $z_\mathcal{V}$ with the input $\mathcal{G}$:

$$\begin{aligned} \mu_\mathcal{V} &= \text{HyperGNN}_\mathcal{V}^\mu(\mathcal{G}), \ \log(\sigma_\mathcal{V}) = \text{HyperGNN}_\mathcal{V}^\sigma(\mathcal{G}), \\ \mu_\mathcal{E} &= \text{HyperGNN}_\mathcal{E}^\mu(\mathcal{G}), \ \log(\sigma_\mathcal{E}) = \text{HyperGNN}_\mathcal{E}^\sigma(\mathcal{G}), \end{aligned} \tag{5}$$

With the learned posterior parameters, then $z_\mathcal{E}$, $z_\mathcal{V}$ can be naturally sampled as follows according to the reparameterization trick [1]: $z_\mathcal{V} = \mu_\mathcal{V} + \sigma_\mathcal{V} \odot \delta$, where $\odot$ is the element-wise product, and $\delta \sim \mathcal{N}(0, I)$ is the standard normal variable. Same process can also be applied to sample $z_\mathcal{E}$. Here we adopt SetGNN as the hypergraph neural network. The key design of it is to learn the multiset functions $f_{\mathcal{V}\rightarrow\mathcal{E}}$ and $f_{\mathcal{E}\rightarrow\mathcal{V}}$. Here $f_{\mathcal{V}\rightarrow\mathcal{E}(\mathcal{E}\rightarrow\mathcal{V})}$ is propagation function to aggregate the vertex (hyperedge) embedding into the hyperedge (vertex) representation. In practice, to facilitate this learning process, we parameterize the multiset functions with the universal multilayer perceptrons (MLP).

**Decoder** With the learned vertex and hyperedge variational distributions, the hypergraph can be reconstructed and generated through the decoding process illustrated in Section 3.3.1 in main text.

### B.4 Limitation

In the paper, we study the problem of how to construct contrastive views of hypergraphs via augmentations, and we are the first to propose hypergraph generative models to generate augmented views, as well as an end-to-end differentiable pipeline to jointly perform hypergraph augmentation and contrastive learning. The extensive experiments demonstrate the effectiveness of our approach. However, since generative augmentation (A6) needs to train the generator model, the substantial improvement of performance comes with the price exhibited by the additional computational cost, this limitation of which we intend to overcome in the future. We show the time consumption of each augmentation method for each training step in Table 3. We can observe that HyperGCL is able to be trained with a reasonable time budget and the running time is roughly linearly proportional to the vertex number. This indicates our methods can scale up. Moreover, how to design more powerful hypergraph generator to fully explore the potential of the hypergraph also remains an open problem.

Table 3: The time consumption of each augmentation method.

| Step/s | Cora | Pubmed | ModelNet40 |
|--------|-------|--------|------------|
| A0 | 0.016 | 0.102 | 0.078 |
| A1 | 0.023 | 0.128 | 0.136 |
| A2 | 0.018 | 0.107 | 0.098 |
| A3 | 0.026 | 0.133 | 0.107 |
| A4 | 0.016 | 0.104 | 0.082 |
| A5 | 0.025 | 0.131 | 0.114 |
| A6 | 0.034 | 0.189 | 0.296 |

### B.5 Potential Social Impact

We are not aware of any potential negative societal impacts regarding our work to the best of our knowledge. For all the used data sets, there is no private personally identifiable information or offensive content.

## C  Implementation Details

### C.1  Experimental Setup

We use Pytorch [2] to build our model. All our experiments were executed on the Linux machine with NVIDIA Tesla V100S GPUs of 32G memory. We tune the hidden dimension of the hypergraph neural network over {64, 128, 256, 512}. The learning rate is searched over {0.1, 0.01, 0.001}, and the weight decay is tuned over {0, 0.00001}. Adam [3] algorithm is used to optimize the model. We set both the number of vertex-to-hyperedge and hyperedge-to-vertex propagation layers to 1, which is suggested by the authors [4]. The number of training epochs is set to 500 by default. For extremely large hypergraphs such as Yelp, for training efficiently and saving computing resources, we extract a child hypergraph of vertex size 16384 to calculate the contrastive loss. Specifically, we randomly sample vertexes of a certain number and all the hyperedges that include those vertexes to construct a child hypergraph. The tradeoff factor $\beta$ in the generator loss is tuned over {1, 2, 5, 10}. The multi-task training tradeoff factor $\alpha$ is searched in the range of {0.1, 0.2, 0.5, 1.0}. For baseline models Self and Con, which also leverage multi-task learning, the tradeoff hyperparameter is tuned in the same range.

### C.2  Data Sets Description

Table 4: Full data set statistics: $|e|$ refers to the size of the hyperedges while $d_v$ refers to the vertex degree.

|  | Cora | Citeseer | Pubmed | Cora-CA | DBLP-CA | Zoo | 20News | Mushroom | NTU2012 | ModelNet40 | Yelp | House | Walmart |
|---|---|---|---|---|---|---|---|---|---|---|---|---|---|
| $|\mathcal{V}|$ | 2708 | 3312 | 19717 | 2708 | 41302 | 101 | 16242 | 8124 | 2012 | 12311 | 50758 | 1290 | 88860 |
| $|\mathcal{E}|$ | 1579 | 1079 | 7963 | 1072 | 22363 | 43 | 100 | 298 | 2012 | 12311 | 679302 | 341 | 69906 |
| # feature | 1433 | 3703 | 500 | 1433 | 1425 | 16 | 100 | 22 | 100 | 100 | 1862 | 100 | 100 |
| # class | 7 | 6 | 3 | 7 | 6 | 7 | 4 | 2 | 67 | 40 | 9 | 2 | 11 |
| max $|e|$ | 5 | 26 | 171 | 43 | 202 | 93 | 2241 | 1808 | 5 | 5 | 2838 | 81 | 25 |
| min $|e|$ | 2 | 2 | 2 | 2 | 2 | 1 | 29 | 1 | 5 | 5 | 2 | 1 | 2 |
| avg $|e|$ | 3.03 | 3.2 | 4.35 | 4.28 | 4.45 | 39.93 | 654.51 | 136.31 | 5 | 5 | 6.66 | 34.72 | 6.59 |
| med $|e|$ | 3 | 2 | 3 | 3 | 3 | 40 | 537 | 72 | 5 | 5 | 3 | 40 | 5 |
| max $d_v$ | 145 | 88 | 99 | 23 | 18 | 17 | 44 | 5 | 19 | 30 | 7855 | 44 | 5733 |
| min $d_v$ | 0 | 0 | 0 | 0 | 1 | 17 | 1 | 5 | 1 | 1 | 1 | 0 | 0 |
| avg $d_v$ | 1.77 | 1.04 | 1.76 | 1.69 | 2.41 | 17 | 4.03 | 5 | 5 | 5 | 89.12 | 9.18 | 5.18 |
| med $d_v$ | 1 | 0 | 0 | 2 | 2 | 17 | 5 | 5 | 5 | 4 | 35 | 7 | 2 |
| $h_e$ | 0.86 | 0.83 | 0.88 | 0.88 | 0.93 | 0.66 | 0.73 | 0.96 | 0.87 | 0.92 | 0.57 | 0.58 | 0.75 |
| $h_v$ | 0.84 | 0.78 | 0.79 | 0.79 | 0.88 | 0.35 | 0.49 | 0.87 | 0.81 | 0.88 | 0.26 | 0.52 | 0.55 |

We use all the thirteen available data sets from the existing hypergraph neural networks literature. The comprehensive benchmark data sets include cocitation networks Cora, Citeseer, Pubmed, and coauthorship networks Cora-CA and DBLP-CA, which are all from [5]. The above data sets have also been used for the study of conventional graph neural networks. But in the hypergraph domain, the construction is changed to better study the nature of hypergraph. In these data sets, the cocitation and coauthor relationships are regarded as the hyperedges. For example, in the hypergraph data set Cora, if more than two articles are cited by another paper, then these cited articles are used to construct a hyperedge. Moreover, we test data sets from the UCI Categorical Machine Learning Repository [6], including 20Newsgroups, Mushroom, and Zoo. Data sets from computer vision and computer graphics domains including ModelNet40 [7] and NTU2012 [8] are also covered. The other three data sets are Yelp, House, and Walmart, which are proposed in [4]. The hypergraph construction methods are described in detail in [9, 4]. For House and Walmart data sets, there are no original vertex features. As suggested by the previous work [4], we use the Gaussian random vectors as the features. One-hot encodings of the labels with added Gaussian noise are used for the vertex features. A hyperparameter is used to control the standard deviation of the Gaussian random vectors. In the experiments, House (0.6) indicates the standard deviation of the random feature is 0.6. The full statistics of these data sets are shown in Table 4.

To better analyze the experimental results, we also design two new metrics $h_e$ and $h_v$ to measure the homophily of the hypergraph from the hyperedge and the vertex perspectives for the first time. $h_e$ is defined as the maximum ratio of vertexes in each hyperedge that have the same label:

$$h_e = \frac{1}{|\mathcal{E}|} \sum_{e \in \mathcal{E}} \max_{c \in C} \left\{ \frac{|\{v : v \in e \wedge y_v = c\}|}{|e|} \right\}, \tag{6}$$

where $C$ is the set of all vertex classes, $|\cdot|$ denotes the number of elements in the set, and $y_v$ is the label of vertex $v$.

$h_v$ calculates the hypergraph homophily from a vertex perspective. We use $\mathcal{E}_v$ to represent all the hyperedges that include vertex $v$. Then all the vertexes in $\mathcal{E}_v$ except $v$ itself are denoted as $v$'s connected set $\mathcal{V}_v$. $h_v$ is then defined as the label consistent ratio between each vertex $v$ and its connected vertex set $\mathcal{V}_v$:

$$h_v = \frac{1}{|\mathcal{V}|} \sum_{v \in \mathcal{V}} \frac{|\{v_i : v_i \in \mathcal{V}_v \wedge y_{v_i} = y_v\}|}{|\mathcal{V}_v|}, \tag{7}$$

## C.3 Data Sets for Fairness

Table 5: The statistics of data sets for fairness: the sensitive feature denotes the feature used to calculate fairness metrics.

|  | German Credit | Recidivism | Credit defaulter |
|---|---|---|---|
| $|\mathcal{V}|$ | 1000 | 18876 | 30000 |
| $|\mathcal{E}|$ | 1000 | 18876 | 30000 |
| Sensitive feature | Gender | Race | Age |
| # feature | 27 | 18 | 13 |
| # class | 2 | 2 | 2 |
| $h_e$ | 0.78 | 0.94 | 0.82 |
| $h_v$ | 0.70 | 0.73 | 0.72 |

To study fair hypergraph representation, we construct three new hypergraph data sets: German [10], Recidivism [11] and Credit [12]. The German data set is from the UCI machine learning category [6] and has 1,000 clients in a German bank. Gender is regarded as the sensitive attribute. The Recidivism data set has 18,876 defendants who got released on bail at the U.S state courts from 1990 to 2009. The race of individuals is regarded as the sensitive attribute. The Credit defaulter hypergraph has 30,000 credit card users. The age of the users is considered as the protected (sensitive) attribute. These three data sets are all binary classification tasks, and the sensitive features are also binary. The hypergraph construction follows the setting in [9]. The top similar individuals of each person in the data set are built as the hyperedges. The similarities are calculated based on the features in the data sets. For measuring fairness, we adopt statistical parity $\Delta_{SP}$ and equalized odds $\Delta_{EO}$. In the hypergraph setting, they measure whether the predictions of each vertex will be influenced by the sensitive attribute. For example, for the Recidivism data set, $\Delta_{SP}$ is defined as the predicted crime probability difference of individuals with different races, and $\Delta_{EO}$ measures the prediction difference by further conditioning on the ground truth crime status y. Therefore, these two metrics can well reflect fairness of the hypergraph model.

Statistical parity [13] is defined as the predicted probability difference of individuals with different protected attributes. It can be written as:

$$\Delta_{SP} = |P(\hat{y}_v = 1 \mid s = 0) - P(\hat{y}_v = 1 \mid s = 1)|, \tag{8}$$

where $\hat{y}_v$ is the vertex predicted label, $s$ is the binary sensitive attribute such as gender. The probabilities are estimated on the test set as in [14]. For equal opportunity [15], it is an alternative criterion by conditioning the fairness metric on the ground truth $y$. It can be formulated as:

$$\Delta_{EO} = |P(\hat{y}_v = 1 \mid y_v = 1, s = 0) - P(\hat{y}_v = 1 \mid y_v = 1, s = 1)| \tag{9}$$

where $y_v$ indicates the vertex ground truth label. The probabilities will also be estimated on the test data set.

## C.4 Robustness Results

In the Table 6, we show more results of baseline methods with regard to robustness under attacks. From the table, we can observe that generalized hyperedge perturbation (A2) still performs the best among fabricated augmentations and our generative augmentation (A6) can outperform these baselines and substantially improve the robustness of HyperGNN.

Table 6: Results on the test data sets with regard to robustness. Bold values indicate the best result. 10% of all vertexes are used for training.

| | Cora | | | Citeseer | | | ModelNet40 | | |
|---|---|---|---|---|---|---|---|---|---|
| | Random | Net | Minmax | Random | Net | Minmax | Random | Net | Minmax |
| SetGNN | 66.87 ± 1.33 | 66.26 ± 1.54 | 66.58 ± 1.02 | 62.89 ± 1.57 | 62.81 ± 1.32 | 62.21 ± 1.64 | 95.74 ± 0.22 | 95.41 ± 0.28 | 93.33 ± 0.26 |
| A1 | 71.30 ± 1.21 | 70.12 ± 1.33 | 70.21 ± 1.04 | 65.94 ± 1.38 | 64.32 ± 1.71 | 63.25 ± 1.81 | 95.81 ± 0.19 | 95.36 ± 0.24 | 93.21 ± 0.21 |
| A2 | 71.90 ± 1.63 | 71.16 ± 0.92 | 70.86 ± 1.22 | 66.41 ± 1.08 | 65.38 ± 1.47 | 64.69 ± 0.98 | 96.09 ± 0.17 | 95.52 ± 0.24 | 93.64 ± 0.26 |
| A3 | 71.39 ± 1.33 | 71.04 ± 1.57 | 70.57 ± 1.04 | 66.12 ± 1.47 | 64.65 ± 1.53 | 64.68 ± 1.23 | 95.81 ± 0.37 | 95.32 ± 0.24 | 93.17 ± 0.18 |
| A4 | 72.11 ± 1.60 | 70.49 ± 1.29 | 70.52 ± 1.39 | 65.94 ± 1.24 | 65.15 ± 1.70 | 64.12 ± 1.19 | 95.79 ± 0.27 | 95.44 ± 0.25 | 93.35 ± 0.24 |
| A5 | 68.87 ± 1.33 | 69.06 ± 1.67 | 68.85 ± 1.12 | 63.74 ± 1.52 | 63.74 ± 1.32 | 63.52 ± 1.63 | 96.05 ± 0.26 | 95.19 ± 0.29 | 93.27 ± 0.35 |
| A6 | **72.15 ± 1.70** | **71.94 ± 1.48** | **71.98 ± 1.36** | **66.60 ± 1.61** | **65.68 ± 1.09** | **65.51 ± 1.13** | **96.58 ± 0.24** | **96.23 ± 0.23** | **94.82 ± 0.33** |
| | NTU2012 | | | House (0.6) | | | House (1.0) | | |
| | Random | Net | Minmax | Random | Net | Minmax | Random | Net | Minmax |
| SetGNN | 73.84 ± 2.18 | 73.38 ± 1.36 | 70.71 ± 1.89 | 67.16 ± 2.55 | 68.88 ± 2.68 | 64.78 ± 2.20 | 56.86 ± 1.93 | 59.95 ± 1.92 | 56.52 ± 2.52 |
| A1 | 74.01 ± 1.76 | 73.44 ± 1.79 | 70.88 ± 1.74 | 67.24 ± 2.78 | 69.14 ± 2.62 | 64.62 ± 2.14 | 57.45 ± 1.85 | 60.52 ± 1.85 | 56.92 ± 2.45 |
| A2 | 74.50 ± 2.03 | 73.86 ± 1.84 | 71.40 ± 1.64 | 67.71 ± 2.94 | 69.59 ± 2.32 | 65.23 ± 2.89 | 57.74 ± 2.70 | 60.73 ± 2.30 | 57.00 ± 1.94 |
| A3 | 73.98 ± 1.74 | 73.45 ± 1.67 | 71.17 ± 1.68 | 67.59 ± 2.05 | 69.23 ± 2.53 | 65.03 ± 2.19 | 57.13 ± 1.60 | 60.42 ± 1.64 | 56.61 ± 2.27 |
| A4 | 73.73 ± 1.59 | 73.72 ± 1.59 | 71.06 ± 1.53 | 67.55 ± 2.41 | 68.85 ± 1.38 | 64.97 ± 3.35 | 57.47 ± 2.72 | 60.10 ± 1.74 | 56.65 ± 2.26 |
| A5 | 74.14 ± 1.68 | 73.52 ± 1.42 | 71.23 ± 1.54 | 67.48 ± 2.26 | 69.11 ± 2.79 | 64.63 ± 2.30 | 56.98 ± 1.59 | 59.88 ± 1.81 | 56.29 ± 2.56 |
| A6 | **75.06 ± 1.97** | **74.37 ± 1.99** | **72.09 ± 1.98** | **69.88 ± 3.27** | **73.14 ± 2.71** | **68.84 ± 2.71** | **60.06 ± 2.07** | **62.41 ± 1.77** | **58.76 ± 2.24** |

Figure 1: Visualization of A3-A5 augmentations. A3 drop vertexes, A4 mask the feature, while A5 preserves partial local structure of the original hypergraph by generating subgraph via random walk.

## C.5 Experimental Effectiveness

Here we show additional experiments to verify the effectiveness of our proposed method. First we show the experimental results in Table 7 with full training data. We used 80% data for training and 20% for validation/testing. We can see the proposed pipeline is still able to achieve improvements though with lower margins.

Then we test the generalization of our proposed method on an additional hypergraph learning task - hypergraph link prediction. The results are summarized in Table 8. We remove 20% relations in the hypergraph and train the model to complete them, AUC is used as the evaluation metric [16]. We experiment with generative augmentation (A6) with the best fabricated augmentation A2 on SetGNN. We can observe that A6 can also achieve substantial improvements in the link prediction setting, which demonstrates the generalization ability of our proposed algorithm.

Moreover, we test with other types of contrastive loss. We adopt the widely used InfoNCE loss in the paper. Here we test our methods with Jensen-Shannon Divergence (JSD) [17] and Triplet Margin (TM) contrastive losses [18]. The experimental results are summarized in the Table 9. The observation is InfoNCE loss works the best in general, and the proposed generative augmentation (A6) is robust to different contrastive objectives.

In addition, we also provide more comparisons with the contrative learning methods on the conventional graph. We include the representative graph contrastive learning method GraphCL [19], and two SOTA graph contrastive learning approaches JOAOv2 [20] and AD-GCL [21]. From the experimental results in Table 10, our proposed generative augmentation (A6) is able to substantially outperform these baselines, which demonstrates the necessity of leveraging higher-order information.

At last, we provide the experiments on different contrast modes. As our main experimental task is node classification, and previous work [18] has empirically verified the node-level contrast manner is better suited to the node-level task. Therefore, following previous works [18, 22], we adopt the widely used node-to-node contrast manner in the main text. In Table 11, we also test the proposed method with the node-to-graph contrast mode. The experimental results verified the benefits of using node-to-node contrastive mode for the node-level task.

Table 7: Results with 80% training data.

|        | Cora | CiteSeer | ModelNet40 | Pubmed |
|--------|------|----------|------------|--------|
| SetGNN | 78.38±2.29 | 71.75±2.90 | 97.66±0.39 | 88.67±0.64 |
| A2     | 79.06±2.57 | 72.35±2.48 | 97.66±0.41 | 89.02±0.74 |
| A6     | **79.65±2.15** | **72.94±2.38** | **98.19±0.40** | **89.34±0.75** |

Table 8: Results on hypergraph link prediction.

|        | Cora | CiteSeer | ModelNet40 | Pubmed |
|--------|------|----------|------------|--------|
| SetGNN | 86.99±1.02 | 84.52±4.87 | 97.71±0.10 | 94.71±0.50 |
| A2     | 87.83±0.82 | 86.19±3.53 | 97.75±0.07 | 94.62±0.40 |
| A6     | **88.69±1.24** | **86.54±4.27** | **97.88±0.09** | **94.83±0.32** |

Table 9: Results with different contrast losses.

|    | InfoNCE | | | | JSD | | | | TM | | | |
|----|------|----------|------------|--------|------|----------|------------|--------|------|----------|------------|--------|
|    | Cora | CiteSeer | ModelNet40 | Pubmed | Cora | CiteSeer | ModelNet40 | Pubmed | Cora | CiteSeer | ModelNet40 | Pubmed |
| A2 | 72.58±1.09 | 66.40±1.35 | 96.56±0.34 | 85.16±0.38 | 70.00±1.32 | 65.40±1.13 | 95.33±0.29 | 84.61±0.51 | 71.67±1.45 | 66.52±1.43 | 95.62±0.27 | 85.28±0.44 |
| A6 | **73.12±1.48** | **66.94±1.00** | **96.93±0.33** | **85.72±0.38** | **72.05±1.24** | **67.02±1.00** | **95.89±0.28** | **85.31±0.40** | **72.12±1.15** | **66.93±1.35** | **96.01±0.23** | **86.00±0.43** |

Table 10: Results with baselines on the conventional graph.

|        | Cora | CiteSeer | Pubmed | ModelNet40 | Yelp |
|--------|------|----------|--------|------------|------|
| SetGNN | 67.93±1.27 | 63.53±1.32 | 84.33±0.36 | 95.85±0.38 | 28.78±1.51 |
| GraphCL | 72.05±1.34 | 65.98±1.43 | 85.16±0.38 | 96.23±0.37 | 31.42±1.25 |
| AD-GCL | 72.36±1.81 | 66.36±1.25 | 84.89±0.44 | 96.11±0.28 | 32.13±1.37 |
| JOAOv2 | 72.23±1.08 | 66.42±1.48 | 85.18±0.32 | 96.34±0.29 | 31.85±0.89 |
| A6     | **73.12±1.48** | **66.94±1.00** | **85.72±0.38** | **96.93±0.33** | **34.64±0.39** |

## C.6 Method Description

In this subsection, we provide more descriptions of the compared augmentation methods and baselines. For the compared baselines, there are existing hypergraph self-supervised approaches Self [23] and Con [24] in recommender systems, and we adapt them in our setting. They both conduct self-supervised learning between the hypergraph and conventional clique-expansion graph, while our augmentation is performed directly on the hypergraph. Self leverage binary cross entropy loss and doesn't perform augmentations on the hyperragph, while Con use InfoNCE loss as the self-supervised objective and conduct random augmentation on the hypergraph and conventional graph.

For the compared augmentation methods, they cover two important perspectives of the hypergraph: structural exploration (A1, A2, A3, A5) and feature perturbation (A4). A0 is a simple identity function that doesn't change the structure or feature of the hypergraph. It's used as the basic augmentation baseline. The correlation between different methods will be shown in the detailed analysis. A1 will drop a certain percentage of hyperedges in the hypergraph, while A2 will drop part of the connections in the converted bipartite graph. A1 can be regarded as a special case of A2 that removes all the connections of the selected hyperedges. A3 alternatively explores the structure from the vertex perspective, which drops the vertexes in the hypergraph. For A5, we perform the random walk to extract a subgraph to explore the local structure of the hypergraph as the augmentation. In addition, we design A4 to explore the effect of feature perturbation. We also visualize A3-A5 to better depict them in Figure 1. These fabricated augmentations well cover the information in the hypergraph from diverse perspectives.

## C.7 Training Curve

In this subsection, we provide the training curve of the generative augmentation A6 compared with the best performing fabricated augmentation A2 and SetGNN. The experimental results of test accuracy with regard to training epochs on Cora and ModelNet40 data sets are shown in Figure 2. From the figure, we can find the training curve of our proposed generative augmentation is very stable and converge very fast, which indicates the training difficulty of A6 will not increase. Moreover, we visualize the training curve with regard to the hyper-parameters $\alpha$ and $\beta$. The number in the

Table 11: Results with different contrastive modes.

|       | Cora         | CiteSeer     | Pubmed       | ModelNet40   |
| ----- | ------------ | ------------ | ------------ | ------------ |
| N2N   | **73.12±1.48** | **66.94±1.00** | **85.72±0.38** | **96.93±0.33** |
| N2G   | 71.28±1.34   | 65.12±1.45   | 84.56±0.41   | 96.15±0.45   |

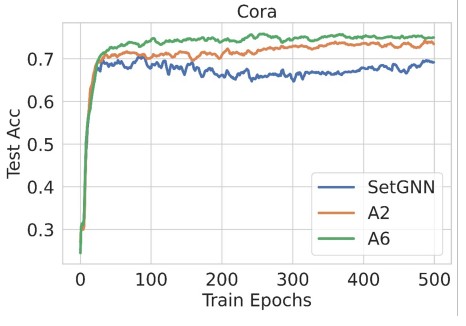 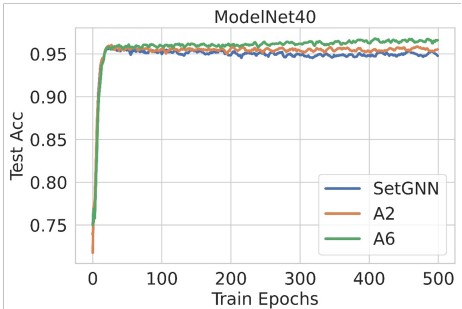

Figure 2: Training Curve on Cora and ModelNet40.

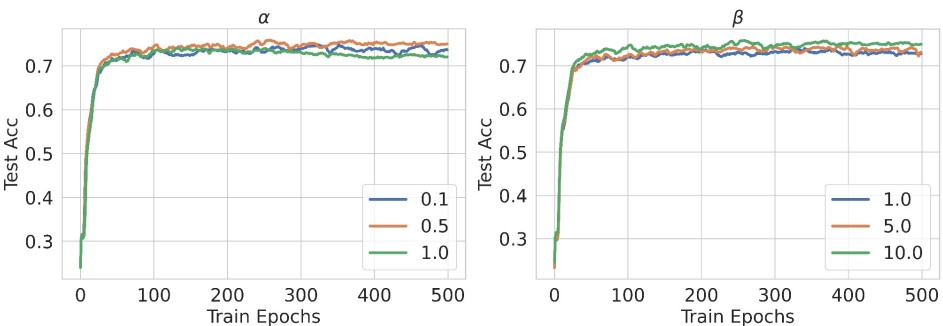

Figure 3: Training Curve on Cora w.r.t hyper-parameters $\alpha$ and $\beta$.

legend represents the value of the hyper-parameter. The results indicate the training of our proposed generative method is stable across different hyper-parameter settings.

## C.8 Checklist Questions for Data Sets.

We have appropriately cited all the used data sets for reference. These data sets are all public accessible where the authors have also stated their consent for the non-commercial usage. Moreover, all the data sets we use do not contain any personally identifiable information or offensive content.