# OpenReview forum: "Augmentations in Hypergraph Contrastive Learning: Fabricated and Generative"
_NeurIPS.cc/2022/Conference — NeurIPS 2022 Accept_

### Official Review · Reviewer_1bGo · 2022-06-28

**Rating:** 6
**Confidence:** 2
**Soundness:** 3 good
**Presentation:** 2 fair
**Contribution:** 3 good

**Summary:**

The submission introduces a generative modelling approach to obtain learnable hypergraph augmentations. These augmentations are used to generate views for contrastive learning. Empirically, the benefit of the learned augmentations over several fabricated types of augmentations, as well as over two baselines, is shown. Additional experiments investigate adversarial robustness and fairness aspects.

**Questions:**

- Did you experiment with the augmentations in different settings, e.g., hypergraph classification, link prediction, ...

- How does the method perform in not low-label regimes (e.g. 50% of vertexes as training set? Does it become infeasible due to the high computational cost of VHGAE? Is the benefit over the baselines still as pronounced?

- Could you clarify the difficulty in constructing hypergraph augmentations.
  >(l.33) However, it is non-trivial to build hypergraph views due to their overly complicated topology, i.e., there are $\sum_e \binom{N}{e}$ possibilities for one hyperedge on $N$ vertices, versus $\binom{N}{2}$ for one edge in graphs.

  At first sight, this property seems to facilitate the construction of hypergraph augmentations.
  What is the "combinatorial challenge" (l.119?) Does this refer to checking the set of hyperedges for duplicates, i.e. whether two finite sets of vertices are equal / correspond to permuted tuples?

- Could you expand on Eq. (2)? How is the bipartite graph determined by the sample $z_{\mathcal V}$ and $z_{\mathcal E}$. What do $z_e$ and $z_v$ correspond to, how are they connected to $z_\mathcal V$ and $z_\mathcal E$? Similarly, $w_{\mathcal {VE}}$ and $w_{ve}$. What is meant with the "topological" distribution of the bipartite graph?

- Regarding equation (4) and the corresponding text. What is meant with $T(G) \circ G$, i.e., how is the "vector applied to perform augmentation". Is $T(G)$ not only a vector, but also a function?

I-  do not understand Figure 3. What do the green arrows indicate?

- In experiments (e.g. Table 3) what is the difference between SetGNN (the backbone) and A0 (identity augmentations).

**Limitations:**

The (presumably) high computational cost of the method is not adequately addressed, i.e., it is not quantified. A comparison of runtimes between generated augmentations, prefabricated augmentations and no augmentations would provide clarity.

**Strengths And Weaknesses:**

### Strengths

+ Compared to other types of data, (e.g. image data), sensible hypergraph augmentations are not obvious and difficult to design.
As far as I am aware, this is the first work to introduce learnable hypergraph augmentations.

+ While the submission focuses on utilizing such augmentations in a contrastive learning setting, the method to generate the augmentations is not limited to it. Application in other settings seem possible.

+ The method consistently outperforms the baselines on the studied vertex classification tasks. The learned augmentations outperform the fabricated ones.

+ The method is not only checked for classification accuracy, also adversarial robustness and fairness aspects are evaluated.

### Weaknesses

- The method is computational expensive. In consequence, only one augmentation in each pair of views is generated (the other is fabricated, cf. l.192)
This might be the reason why the empirical evaluation focuses on the low-label regime, where fewer augmentations are required.

- Some parts of the paper are difficult to understand, particularly the description of the augmentation procedure. See questions.

- The adversarial attack experiments are not very expressive, considering that for some attacks model accuracy even increases. Maybe the adaptions of Random Noise and Minmax Attack could even serve as hypergraph augmentations.


**Post rebuttal edit**

While my questions have been adequately addressed, several reviewers have pointed at limited novelty in comparison with existing works from the graph augmentation literature. I will thus keep my initial rating of **6: Weak accept**.

---

> ### Author Response · Authors · 2022-08-02
> **Response to Reviewer 1bGo (1/2)**
>
> Thank you for your valuable review and suggestions.
>
> **Q1:** The (presumably) high computational cost of the method is not adequately addressed, i.e., it is not quantified.
>
> **A1:** Thanks for your valuable advice. We’ve examined the computational cost. Please refer to our general comments.
>
> -----------------------
>
> **Q2:** How does the model perform in not low-label regimes? It’s infeasible due to the high computational cost of VHGAE?
>
> **A2:** We choose the low-label regime not because of the computing cost. This is because in the low-label regime, little information can be utilized from labels, and contrastive learning here can leverage the structural information to benefit the model a lot. In the experiments, we can find with label rate increase, the improvement saturated. This is also verified in many previous works about self-supervised learning [1,2,3]. Besides, we also show the experimental performance under 80% training data. We can observe the proposed pipeline is still able to achieve improvements
>
> |     80% Data/Acc    |        Cora       |      CiteSeer     |     ModelNet40    |       Pubmed      |
> |:-------------------:|:-----------------:|:-----------------:|:-----------------:|:-----------------:|
> |        SetGNN       |     78.38±2.29    |     71.75±2.90    |     97.66±0.39    |     88.67±0.64    |
> |          A2         |     79.06±2.57    |     72.35±2.48    |     97.66±0.41    |     89.02±0.74    |
> |          A6         |     79.65±2.15    |     72.94±2.38    |     98.19±0.40    |     89.34±0.75    |
>
> -----------------------
>
> **Q3:** Can adaptations of Random Noise and Minmax Attack even serve as hypergraph augmentations?
>
> **A3:** We thank the reviewer for the insightful suggestion. Following this direction, the immediately challenge is the computational cost, where adversarial attacks (on the discrete space) meet the combinatorial complexity of hypergraph structures (with $N$ vertices, there are $\sum_{e=1}^N \tbinom{N}{e}$ possibilities for one hyperedge, versus $\tbinom{N}{2}$ for one edge in graphs). We are unable to tackle this challenge in the current stage, and believe it is an interesting future direction.
>
> -----------------------
>
> **Q4:** Did you experiment with the augmentations in different settings
>
> **A4:** We’re glad to examine in different settings. We show the results of hypergraph link prediction below (we remove 20% relations in the hypergraph and train the model to complete them, AUC is used as the evaluation metric). We experiment with generative augmentation (A6) with the best fabricated augmentation A2 on SetGNN. We can observe that A6 can also achieve substantial improvements in the link prediction setting, which demonstrates the generalization ability of our proposed algorithm.
>
> |     Link/AUC   |         Cora|      CiteSeer     |     ModelNet40    |       Pubmed      |
> |:---------------:|:-----------------:|:-----------------:|:-----------------:|:-----------------:|
> |      SetGNN     |     86.99±1.02    |     84.52±4.87    |     97.71±0.10    |     94.71±0.50    |
> |        A2       |     87.83±0.82    |     86.19±3.53    |     97.75±0.07    |     94.62±0.40    |
> |        A6       |     88.69±1.24    |     86.54±4.27    |     97.88±0.09    |     94.83±0.32    |
>
> -----------------------
>
> [1] JGCL: Joint Self-Supervised and Supervised Graph Contrastive Learning. WWW 2022
>
> [2] SimGRACE: A Simple Framework for Graph Contrastive Learning without Data Augmentation. WWW 2022
>
> [3] Graph Contrastive Learning with Augmentations. NeurIPS 2020

---

> > ### Author Response · Authors · 2022-08-02
> > **Response to Reviewer 1bGo (2/2)**
> >
> > Thanks again for your helpful feedback. We continue to answer reviewer's questions.
> >
> > **Q5:** Could you clarify the difficulty in constructing hypergraph augmentations
> >
> > **A5:** The combinatorial challenge denotes that it’s hard to enumerate all the possibilities as one hyperedge can consist of any number of vertices. With N vertices, there are $\sum_{e=1}^N (N, e)$ possibilities for one hyperedge, versus $(N, 2)$ for one edge, which means it’s harder on the hypergraph to choose which relation to augment.
> >
> > -----------------------
> >
> > **Q6:** Could you expand on Eq. (2)?
> >
> > **A6:** $\mathcal{V}$ denotes the set of vertices and $\mathcal{E}$ denotes the set of hyperedges. $z_{\mathcal{V}}$ denotes the set of vertex embeddings, while $z_{\mathcal{E}}$ represents the set of hyperedge embeddings. $z_{\mathcal{v}}$ and $z_{\mathcal{e}}$ denote the embeddings of a specific vertex/hyperedge. Similarly, $w_{ve}$ represents the decoded probability of the relation between vertice $v$ and hyperedge $e$. For the decoded topology distribution, we mean that $z_{\mathcal{V}}$ and $z_{\mathcal{V}}$ decode the edge structure in the hypergraph into embeddings.
> >
> > -----------------------
> >
> > **Q7:** What do the green arrows in Figure 3 indicate?
> >
> > **A7:** We apologize for the confusion. The green lines indicate these modules participated in the process of calculating and optimizing the proposed loss function. We clarify accordingly in revision.
> >
> > -----------------------
> >
> > **Q8:** In experiments (e.g. Table 3) what is the difference between SetGNN (the backbone) and A0 (identity augmentations).
> >
> > **A8:** A0 denotes using identity function as the augmentation and optimizing the contrastive loss. It’s used as a simple augmentation baseline. On the other hand, SetGNN only optimizes the labeled classification loss.
> >
> > -----------------------
> >
> > **Q9:** How is the "vector applied to perform augmentation". Is  T(G)  not only a vector, but also a function?
> >
> > **A9:** T(G) is not a vector, but a transform function, i.e A0-A6.

---

> > ### Comment · Reviewer_1bGo · 2022-08-05
> > **Response to authors**
> >
> > I thank the authors for their response. I have two remaining questions.
> >
> > **Q1** Since computational cost does not seem to be an issue, could you put the following into context (l. 191) "Due to the computational cost of VHGAE, we train one VHGAE to produce one generative view, with the other view  ̃Gp is kept as fabricated."
> >
> > **Q2** I still dot understand Eq. (4). From my understanding, $T(G) = \operatorname{Sigmoid}((\mathcal w_{\mathcal {VE}} + \log(\delta) - \log(1-\delta))\tau)$, is the output of the sigmoid activation function on the decoded probability relations and not a transform function. Still, if $T(G)$ was a transform function, why does it depend on $G$?

---

> > > ### Author Response · Authors · 2022-08-07
> > > **Response to Questions**
> > >
> > > Thank you very much for the follow-up questions! We hope that the following answers would address any remaining concerns.
> > >
> > > **A1:** We apologize for the incomplete expression in the original text that led to the confusion. We meant to say in line 191 that, “Due to the computational cost of **collaboratively** optimizing two generative views [1,2], we train one VHGAE to produce one generative view, with the other view prefixed as fabricated.” Here we missed the word “collaborative” which is the cause of the computational overhead when learning two generative hypergraph views. To be specific,
> > >
> > > 	(i) independently optimizing two hypergraph generators is of reasonable budgets but would lead to distribution collapse (i.e., two hypergraph generators output the same distribution) [1,2] which results in less effective generative views (we observe the substantial performance drop e.g. 73.12->71.08 on Cora), while
> > > 	(ii) the collaborative optimization techniques for graph generators (e.g. REINFORCE on the rewards of generative graph structures) are not directly applicable to HyperGCL due to the combinatorial challenge of hypergraph structures (which is computationally expensive).
> > > We therefore only free one view to be explored by generators and leave the collaborative optimization to future works.
> > >
> > > [1] Bringing Your Own View: Graph Contrastive Learning without Prefabricated Data Augmentations. WSDM 2022
> > >
> > > [2] AutoGCL: Automated Graph Contrastive Learning via Learnable View Generators. AAAI 2022
> > >
> > > **A2:** We apologize for the confusing definition of $T(\mathcal{G})$. Eq. (4) ($T(\mathcal{G})$) is the output of Gumbel-Softmax [1,2,3] which stochastically samples relations based on the decoded probabilities. In the equation $\text{Sigmoid}((w_{\mathcal{V}\mathcal{E}}+\text{log}(\delta)-\text{log}(1-\delta))/\tau)$, $\delta\sim \text{Uniform}(0,1)$ is a random variable. This equation leverages the reparameterization tricks [1,2,3] to make the outputs binary (0/1). A value of 1 indicates that the relation is sampled. Since $T(\mathcal{G})$ denotes which relation to augment, we said $T(\mathcal{G})$ could serve the function of transforming the original hypergraph to the generated one. For our proposed generative augmentation, the parameters of the generator are learned from the hypergraph data $\mathcal{G}$. It takes the hypergraph as input and transforms it into the augmented hypergraph. Therefore, we write the transform function as $T(\mathcal{G})$.
> > >
> > >
> > > [1] Categorical Reparameterization with Gumbel-Softmax. ICLR 2017
> > >
> > > [2] The Concrete Distribution: A Continuous Relaxation of Discrete Random Variables. ICLR 2017
> > >
> > > [3] GANS for Sequences of Discrete Elements with the Gumbel-softmax Distribution. arxiv

---

### Official Review · Reviewer_RMPZ · 2022-07-04

**Rating:** 5
**Confidence:** 4
**Soundness:** 3 good
**Presentation:** 3 good
**Contribution:** 3 good

**Summary:**

In the paper, the author studies the problem of how to construct contrastive views of hypergraphs via augmentations. Its contributions include: (1) Fabricating two schemes to augment hyperedges with higher-order relations encoded, and adopt three vertex augmentation strategies from graph-structured data. (2) Proposing hypergraph generative models to generate augmented views, and then an end-to-end differentiable pipeline to jointly perform hypergraph augmentation and contrastive learning.

**Questions:**

About adversarial robustness in experiments: the authors mentioned that " Meanwhile current attack algorithms are far from effective because they are not specifically designed for hypergraphs ." So are the experimental results convincing? About Table7: Why does not the table contain experimental results of A1, A3 and A5?

**Limitations:**

In the experiments, one have to select the data augmentation strengths manually. The fairness of experiments needs more datasets to support.

**Strengths And Weaknesses:**

Strengths:
The quality and clarity of the paper are good. The motivation is clear and the author does enough experiments. The paper is sound in improving the generalizability of hypergraph neural networks in the low-label regime.
Weaknesses:
The innovation is of HyperGraphCL is limited. The augmentations like perturbing hyperedges and subgraph, are just the same as that in graph contrastive learning [1][2][3]. Why does the generative model that the author proposes work well? I think the author needs to explain more clearly in theory. The description and analysis in the experiment section are ambiguous, and the writing in this part is hard to follow. The compared baselines are not advanced enough, and I recommend the author conduct more SOTA methods for comparisons, e.g., SOTA graph contrastive methods.

[1] You, Y., Chen, T., Sui, Y., Chen, T., Wang, Z., & Shen, Y. (2020). Graph contrastive learning with augmentations. Advances in Neural Information Processing Systems, 33, 5812-5823.
[2] Suresh, S., Li, P., Hao, C., & Neville, J. (2021). Adversarial graph augmentation to improve graph contrastive learning. Advances in Neural Information Processing Systems, 34, 15920-15933.
[3] Xia, J., Wu, L., Chen, J., Hu, B., & Li, S. Z. (2022, April). SimGRACE: A Simple Framework for Graph Contrastive Learning without Data Augmentation. In Proceedings of the ACM Web Conference 2022 (pp. 1070-1079).

---

> ### Author Response · Authors · 2022-08-02
> **Response to Reviewer RMPZ**
>
> We appreciate your helpful feedback. We hope the following answers will address your concerns.
>
> **Q1:** Is the innovation of HyperGCL limited?
>
> **A1:** Here we refer reviewer to our general response.
>
> -----------------------
>
> **Q2:** Why does the generative model that the author proposes work well?
>
> **A2:** Although fabricated augmentations are simple and effective, in the augmentation space they might be located within the restricted subspace. Insteadly, generative augmentations are able to explore beyond such subspace in a data-driven manner (i.e. “extrapolation”) [1, 4], searching for better ways to augment (in theory). We here add additional analysis of hypergraph generative augmentations. In Figure 4(b), we visualize two hyperedges before and after the generative augmentation, where extraneous information (related to downstream) is removed with hypergraph homophily improved, which is not achievable with (combined) fabricated augmentations (where entities i.e. nodes/hyperedges are perturbed randomly and uniformly).
>
> -----------------------
>
> **Q3:** Description and analysis in the experiment section are ambiguous.
>
> **A3:** Sorry for the ambiguous description. We’ve modified the text.
>
> -----------------------
>
> **Q4:** The compared baselines are not advanced enough.
>
> **A4:** Per the suggestion, we are glad to provide more comparisons to address the reviewer's concerns. Please see the results as follows. We include the representative graph contrastive learning method GraphCL [1], and two SOTA graph contrastive learning approaches JOAOv2 [2] and AD-GCL [3]. Our proposed generative augmentation (A6) is able to substantially outperform these baselines, which demonstrates the necessity of leveraging higher-order information.
>
> |                |        Cora       |      CiteSeer     |       Pubmed      |     ModelNet40    |        Yelp       |
> |:--------------:|:-----------------:|:-----------------:|:-----------------:|:-----------------:|:-----------------:|
> |      SetGNN    |     67.93±1.27    |     63.53±1.32    |     84.33±0.36    |     95.85±0.38    |     28.78±1.51    |
> |     GraphCL    |     72.05±1.34    |     65.98±1.43    |     85.16±0.38    |     96.23±0.37    |     31.42±1.25    |
> |      AD-GCL    |     72.36±1.81    |     66.36±1.25    |     84.89±0.44    |     96.11±0.28    |     32.13±1.37    |
> |      JOAOv2    |     72.23±1.08    |     66.42±1.48    |     85.18±0.32    |     96.34±0.29    |     31.85±0.89    |
> |        A6      |     73.12±1.48    |     66.94±1.00    |     85.72±0.38    |     96.93±0.33    |     34.64±0.39    |
>
> -----------------------
>
> **Q5:** Are the experimental results on robustness convincing?
>
> **A5:** We apologize for the misleading wording. Here we refer reviewer to the general response.
>
> -----------------------
>
> **Q6:** About Table7: Why does not the table contain experimental results of A1, A3 and A5?
>
> **A6:** We adopt the current three augmentation strategies because these three methods are the most representative. A2 and A4 explores the augmentation strategies from the structure and node feature perspective, while A6 is our proposed generative augmentation. A2 and A6 also work the best among all augmentations. Therefore, we use them as the baselines for Table 7. The other three augmentation baselines are omitted mainly for space complexity. We’re glad to also provide the results of the other three (A1, A3, A5) augmentation strategies. The results are added to Appendix C.4 Table 5. From the table, we can observe that generalized hyperedge perturbation (A2) still performs the best among fabricated augmentations and our generative augmentation (A6) can outperform these baselines and substantially improve the robustness of HyperGNN.
>
> -----------------------
>
> [1] Graph Contrastive Learning with Augmentations. NeurIPS 2020
>
> [2] Graph Contrastive Learning Automated. ICML 2021
>
> [3] Adversarial Graph Augmentation to Improve Graph Contrastive Learning. NeurIPS 2021
>
> [4] Bringing Your Own View: Graph Contrastive Learning without Prefabricated Data Augmentations. WSDM 2022

---

### Official Review · Reviewer_9Xc8 · 2022-07-07

**Rating:** 5
**Confidence:** 4
**Soundness:** 2 fair
**Presentation:** 2 fair
**Contribution:** 2 fair

**Summary:**

The paper proposes augmentations for learning representation on HyperGraphs using contrastive learning. They introduce two classes of augmentations: fabricated and generative augmentations and show that using these augmentations it is possible to learn good representations over the input hypergraphs. The paper is well-written and easy-to-follow.

**Questions:**

1- How does the proposed augmentations interact with other types of contrastive loss?
2- Is it possible to define the augmentations over the hypergraph space rather than converting it to a a bipartite graph?

**Ethics Review Area:**

["I don’t know"]

**Strengths And Weaknesses:**

Strengths:
- The paper tackles a problem (representation learning on HyperGraphs) that can be used in various applications.
- The paper is well-written and easy-to-follow.
- The experimental setup is fair and the model shows good performance compared to baselines.

Weaknesses:
- The paper simply adopts previous work on contrastive graph representation learning [1,2,3] and applies it to hyper-graphs. There's not much insights on hypergraphs but rather authors map hypergraphs into bipartite graphs and apply recent graph augmentations to learn representations. This is not a bad thing per se, but again it doesn't give any new insights about hypergraphs.
- Experimental studies need to be expanded to show for example how the augmentation mechanism works with other losses beside InfoNCE.

[1] https://arxiv.org/abs/2201.09830
[2] https://arxiv.org/abs/2006.06830
[3] https://arxiv.org/abs/2106.07594

---

> ### Author Response · Authors · 2022-08-02
> **Response to Reviewer 9Xc8**
>
> **Q1:** Does the paper simply adopts previous work on contrastive graph representation learning and applies it to hyper-graphs?
>
> **A1:** Here we refer reviewer to our general response.
>
> -----------------------
>
> **Q2:** How does the proposed augmentations interact with other types of contrastive loss?
>
> **A2:** Following previous works [1, 2], we adopt the widely used InfoNCE loss and people in the community usually adopt single loss in their paper. But we’re also glad to test our methods with different contrastive losses. We test our methods with Jensen-Shannon Divergence (JSD) [3] and Triplet Margin (TM) contrastive losses [4]. The experimental results are summarized in the table below. The observation is InfoNCE loss works the best in general, and the proposed generative augmentation (A6) is robust to different contrastive objectives.
>
> |           |       InfoNCE     |         InfoNCE           |     InfoNCE               |           InfoNCE         |         JSD       |          JSD                |       JSD                   |        JSD                  |         TM        |           TM                |         TM                  |       TM                    |
> |:---------:|:-----------------:|:-----------------:|:-----------------:|:-----------------:|:-----------------:|:-----------------:|:-----------------:|:-----------------:|:-----------------:|:-----------------:|:-----------------:|:-----------------:|
> |           |        Cora       |      CiteSeer     |     ModelNet40    |       Pubmed      |        Cora       |      CiteSeer     |     ModelNet40    |       Pubmed      |        Cora       |      CiteSeer     |     ModelNet40    |       Pubmed      |
> |     A2    |     72.58±1.09    |     66.40±1.35    |     96.56±0.34    |     85.16±0.38    |     70.00±1.32    |     65.40±1.13    |     95.33±0.29    |     84.61±0.51    |     71.67±1.45    |     66.52±1.43    |     95.62±0.27    |     85.28±0.44    |
> |     A6    |     73.12±1.48    |     66.94±1.00    |     96.93±0.33    |     85.72±0.38    |     72.05±1.24    |     67.02±1.00    |     95.89±0.28    |     85.31±0.40    |     72.12±1.15    |     66.93±1.35    |     96.01±0.23    |     86.00±0.43    |
>
> -----------------------
>
> [1] Graph Contrastive Learning with Adaptive Augmentation. WWW 2021
>
> [2] SimGRACE: A Simple Framework for Graph Contrastive Learning without Data Augmentation. WWW 2022
>
> [3] Deep Graph Infomax. ICLR 2019
>
> [4] An Empirical Study of Graph Contrastive Learning. NeurIPS 2021

---

### Official Review · Reviewer_7pvs · 2022-07-11

**Rating:** 4
**Confidence:** 3
**Soundness:** 2 fair
**Presentation:** 3 good
**Contribution:** 3 good

**Summary:**

This paper investigates the view augmentations in hypergraph contrastive learning. In particular, it explores several domain knowledge-guided fabrication augmentation methods and proposed hypergraph generative models. The authors' experiments show that generative augmentations perform well at preserving higher-order information and thus presents better performance.

**Questions:**

1. if "current attack algorithms are far from effective because they are not specifically designed for hypergraphs", then why you claim the designed hyperGCL can provide robustness?
2. Why statistical parity ΔSP and equalized odds ΔEO can represent fairness in hypergraph learning?
3. Are the Hypergraph Generative Models hard to train (converge)?

**Ethics Review Area:**

["I don’t know"]

**Limitations:**

Yes

**Strengths And Weaknesses:**

Strengths:
1.The paper is easy to follow.
2. The proposed hypergraph generative modeling is a good strategy for augmenting a view and the results seem quite promising.

Weaknesses:
1. The reason or analysis why the proposed hypergraph generative modeling works better than other view augmenting methods are not clear.
2. I think the hypergraph generative modeling could potentially increase training difficulty compared to other methods.
3. If with different augmentations, the contrast manner should be accordingly tuned?

Others:
I'm not quite sure if the topic (hypergraph contrastive learning) is important or not (I have very bad experience with hypergraph learning, as they cannot work well in real applications).

---

> ### Author Response · Authors · 2022-08-02
> **Response to Reviewer 7pvs**
>
> We appreciate your helpful feedback. We hope the following answers will address your concerns.
>
> **Q1:** Why the proposed hypergraph generative modeling works better?
>
> **A1:** Although fabricated augmentations are simple and effective, they might be located within the restricted augmentation subspace. Insteadly, generative augmentations are able to explore beyond such subspace (i.e. “extrapolation”) in a data-driven manner [11, 12], searching for better ways for augmenting. We here add additional analysis of hypergraph generative augmentations. In Figure 4(b), we visualize two hyperedges before and after the generative augmentation, where extraneous information (related to downstream) is removed with hypergraph homophily improved, which is not achievable with (combined) fabricated augmentations (where entities i.e. nodes/hyperedges are perturbed randomly and uniformly).
>
> -----------------------
>
> **Q2:** Hypergraph generative modeling could potentially increase training difficulty.
>
> **A2:** The training difficulty will not increase. We show the training curve of the generative augmentation A6 compared with the best performing fabricated augmentation A2 and SetGNN. We’ve added it into appendix C.6 (Figure 2). From the figure, we can find the training curve of our proposed generative augmentation is very stable and converges very fast.
>
> -----------------------
>
> **Q3:** Should the contrast manner be accordingly tuned?
>
> **A3:** As our experimental task is node classification, and previous work [10] has empirically verified the node-level contrast manner is better suited to the node-level task. Therefore, following previous works [1,10], we adopt the widely used InfoNCE loss and node-to-node contrast manner. We are glad to provide more results of different manners, while we are unable to finish in time (during rebuttal). We promise to provide such results in the future.
>
> -----------------------
>
> **Q4:** Why you claim the designed hyperGCL can provide robustness?
>
> **A4:** We apologize for the misleading wording. Here we refer reviewer to the general response.
>
> -----------------------
>
> **Q5:** Why statistical parity ΔSP and equalized odds ΔEO can represent fairness in hypergraph learning?
>
> **A5:** Statistical parity and equalized odds [8, 9] are two widely used metrics for measuring fairness. In the hypergraph setting, they measure whether the predictions of each vertex will be influenced by the sensitive attribute. For example, for the Recidivism data set, ΔSP is defined as the predicted crime probability difference of individuals with different races, and ΔEO measures the prediction difference by further conditioning on the ground truth crime status y. Therefore, these two metrics can well reflect fairness of the hypergraph model.
>
> -----------------------
>
> **Q6:** Is the topic (hypergraph contrastive learning) important or not?
>
> **A6:** The topic is important and general. Hypergraph is a critical tool to represent the complex relationships between objects. Hypergraph neural networks are developing and improving all the time, and it has been successfully applied to recommender systems [2,3], financial analyses [4], bioinformatics [5], text analyses [6] and object classification [7].
>
> -----------------------
>
> [1] Graph Contrastive Learning with Adaptive Augmentation. WWW 2021
>
> [2] Self-Supervised Hypergraph Convolutional Networks for Session-based Recommendation. AAAI 2021
>
> [3] Double-Scale Self-Supervised Hypergraph Learning for Group Recommendation. CIKM 2021
>
> [4] Spatiotemporal Hypergraph Convolution Network for Stock Movement Forecasting. ICDM 2020
>
> [5] Multiscale and Integrative Single-cell Hi-c Analysis with Higashi. Nature Biotechnology
>
> [6] Be More with Less: Hypergraph Attention Networks for Inductive Text Classification. EMNLP 2020
>
> [7] Hypergraph Neural Networks. AAAI 2019
>
> [8] Equality of opportunity in supervised learning. NeurIPS 2016
>
> [9] Learning Adversarially Fair and Transferable Representations. ICML 2018
>
> [10] An Empirical Study of Graph Contrastive Learning. NeurIPS 2021
>
> [11] Graph Contrastive Learning Automated. ICML 2021
>
> [12] Bringing Your Own View: Graph Contrastive Learning without Prefabricated Data Augmentations. WSDM 2022

---

> > ### Comment · Reviewer_7pvs · 2022-08-08
> > **follow-up question**
> >
> > Thanks for the authors' response. I'm fine with most of their answers. I have a question about the answers to Q2. The figure indeed shows that the training is stable, however, is that a result of carefully tuning hyper-parameters?

---

> > > ### Author Response · Authors · 2022-08-08
> > > **Response to Follow-up Questions**
> > >
> > > Thanks very much for your response. Our proposed method is stable and the figure is not a result of carefully tuning hyper-parameters. The stability of our method w.r.t hyper-parameters is similar to the generalization performance shown in Tables 1 and 2 in Appendix, which is not sensitive to hyper-parameters. We'll put figures of multiple sets of hyper-parameters in the final version of our paper.

---

> > ### Author Response · Authors · 2022-08-09
> > **Gentle Reminder**
> >
> > Dear reviewer 7pvs,
> >
> > We sincerely appreciate your follow-up discussion with us. We hope that you have found our clarification clear. We would be thrilled to discuss more to address your remaining concerns, or if you have already found our response satisfactory, we humbly remind you of a fitting update of the rating. Thank you again for your time and efforts!
> >
> > Sincerely,
> > All Authors

---

### Official Review · Reviewer_V7YT · 2022-07-11

**Rating:** 6
**Confidence:** 3
**Soundness:** 3 good
**Presentation:** 3 good
**Contribution:** 3 good

**Summary:**

The authors present a novel approach to perform contrastive learning by investigating several fabricated view augmentations of the hypergraph and a variational generative model. Their exhausting experiments demonstrated that the proposed methodology results in the SOTA for hypergraph, especially in low data regimens.

**Questions:**

1. Section 3.2, when describing the different fabricated augmentations. Although it is clear that augmentations A3-A5 come from previous work, it would be nice to have a figure where A3-A5 are depicted
2. Line 191-192. ‘We train one VHGAE to produce one generative view, with the other view G ̃p is kept as prefabricated.’ Do they mean the other view? Otherwise, do you generate one view from another view? Please explain
3. Table 2 is never referred to in the text.
4. Since Self and Con are the methods, which are compared, it would be a good idea to include some information about these approaches in the related work.

**Limitations:**

Authors have not adequately commented on the known limitations.

**Strengths And Weaknesses:**

Strengths:

1. Well written, good flow of ideas.
2. Extensive evaluation:
    1. 13 datasets with different data regimens 10%, 1%.
    2. Different self-supervision mechanisms
    3. Comparison with approaches to convert the hypergraph to graph
3. Novel idea to construct augmented views of hypergraphs for contrastive learning, especially the generative model

Weaknesses:

1. One of the arguments all around the paper is that the proposed method works for a low data regime. However, the authors performed experiments with 10% and 1%. What do the results look like when using ˜100% data for training? In other words, could you show the upper bound?

2. The results from the robustness part of the experiments are inconclusive. The authors explain that in some cases, the performance even increased. Given that these perturbations on the hypergraphs were adopted for attacks on normal graphs. The authors could either provide some insights into their results or perform the robustness study using suitable hypergraph perturbations

---

> ### Author Response · Authors · 2022-08-02
> **Response to Reviewer V7YT**
>
> We thank you for your review and appreciate all your helpful feedback.
>
> **Q1:** What do the results look like when using ˜100% data for training? In other words, could you show the upper bound?
>
> **A1:** We show the results with 80% data for training and 20% for validation/test. We can see the proposed pipeline is still able to achieve improvements though with lower margins (versus low-data).
>
> |     80% Data/Acc    |        Cora       |      CiteSeer     |     ModelNet40    |       Pubmed      |
> |:-------------------:|:-----------------:|:-----------------:|:-----------------:|:-----------------:|
> |        SetGNN       |     78.38±2.29    |     71.75±2.90    |     97.66±0.39    |     88.67±0.64    |
> |          A2         |     79.06±2.57    |     72.35±2.48    |     97.66±0.41    |     89.02±0.74    |
> |          A6         |     79.65±2.15    |     72.94±2.38    |     98.19±0.40    |     89.34±0.75    |
>
> -----------------------
>
> **Q2:** The results from the robustness part of the experiments are inconclusive.
>
> **A2** We apologize for the misleading wording.  We here refer the reviewer to the general response.
>
> -----------------------
>
> **Q3:** Although it is clear that augmentations A3-A5 come from previous work, it would be nice to have a figure where A3-A5 are depicted.
>
> **A3:** We’ve added detailed descriptions in Appendix C.5 about different augmentations. We also visualize A3-A5 in Figure 1 in Appendix to make it more clear.
>
> -----------------------
>
> **Q4:** For the sentence 'The other view $\tilde{\mathcal{G}}_p$ is kept as prefabricated.’ Do they mean the other view?
>
> **A4:** We apologize for the misleading description. For line 191-192, yes, they mean the other view. We don’t generate one view from another view.
>
> -----------------------
>
> **Q5:** Table 2 is never referred to in the text.
>
> **A5:** Thanks for pointing it out. We’ve revised the text accordingly.
>
> -----------------------
>
> **Q6:** Include some information about these approaches in the related work.
>
> **A6:** Thanks very much for your advice. We’ve added some descriptions of these baseline methods in Appendix C.5.

---

> ### Comment · Reviewer_V7YT · 2022-08-08
> **Response to Conference Paper4427 Authors**
>
> I thank the authors for the response. Clearly, increasing the sample size in the dataset narrows down the performance gain. Further, thanks for clarifying the robustness issue. Hence, I stick to my original rating.

---

> > ### Author Response · Authors · 2022-08-08
> > **Response to Reviewer V7YT**
> >
> > Thanks very much for your response. We are glad that our clarification is clear to resolve the concerns. We sincerely appreciate the reviewer’s time and efforts.

---

### Official Review · Reviewer_jPDp · 2022-07-15

**Rating:** 6
**Confidence:** 5
**Soundness:** 3 good
**Presentation:** 3 good
**Contribution:** 2 fair

**Summary:**

This paper aims to improve the generalizability of hypergraph neural networks using self-supervised contrastive learning in a few-shot learning scenario. The paper's main contributions can be summarized as follows: first, using domain knowledge, they fabricated augmentations, and second, for the first time, they proposed hypergraph generative models to generate augmented views using contrastive learning and novel variational hypergraph auto-encoder architecture which is an adaptation of VAE.

**Questions:**


1. Are these augmentations extensive enough or is there any correlation between them?

2. Can you provide information about computational costs?

3. Are generated augmentations, combinations of the fabricated augmentations? Can you provide further analysis?

4. Eq5: I believe you need a “max” rather than a “min”, as ELBO is to be maximized.

5. Please explain briefly L_cl as the contrastive learning loss after the corresponding equation in the main text, and say further details are in the Appendix.


**Limitations:**

The authors stated two limitations of the study: the additional computational cost due to hypergraph generative model, and how to design powerful hypergraph generators, in the Appendix B4.
In Appendix B5, no negative societal impact is reported. When the application domain includes Recommender systems, I believe society is always at risk under harmful/unfair content spread/recommendation due to current ML/DL methods. This should be acknowledged.

**Strengths And Weaknesses:**

Strengths:
* Paper is well written, flows smoothly, and easy to understand.
* Even though the idea is a combination of well-known techniques, overall it is quite original.
* The proposed method is tested with a high number of datasets and experiments even for robustness and fairness. I believe the experimental sections of the paper are quite strong but could be improved (see weaknesses).

Weaknesses:
* Proposed augmentations from A0 to A5 were not investigated in detail. Are these augmentations independent or is there any correlation between them? The selection of augmentations and coverage of the augmentations should be further investigated. Furthermore, there are not any combinations such as A1 and A2 together. This lack of various combinations of augmentations may be the reason for the proposed method to be more successful, this is not clear.
* Computational cost analysis of the proposed model is missing.
* Generated augmentations are not investigated enough. Are these augmentations, combinations of the fabricated augmentations?

---

> ### Author Response · Authors · 2022-08-02
> **Response to Reviewer jPDp**
>
> We appreciate your helpful feedback. We hope the following answers will address your concerns.
>
> **Q1:** Are these augmentations extensive enough or is there any correlation between them?
>
> **A1:** The proposed augmentations are extensive and cover two important perspectives of the hypergraph: structure (A1,A2, A3, A5) and node feature (A4) augmentations. Correlation among augmentations: Within structure augmentations, A1 is operated on the original hyper-structure and A2 on the equivalent view of the transformed bipartite graph. A3 conducts vertex dropping globally uniformly, while A5 considers the local subgraph pattern. Within node feature augmentation, A4 conducts feature masking to explore the effect of features in the hypergraph. Lastly, A6 is the augmentation learned in a data-driven manner rather than fabricated. We believe the proposed augmentations are representative to generate perturbations in diverse aspects on hypergraphs, and we would also humbly welcome any suggestions from reviewer to further explore the augmentation space.
>
> -----------------------
>
> **Q2:** Can you provide information about computational costs?
>
> **A2:** Here we refer reviewer to the general response.
>
> -----------------------
>
> **Q3:** Are generated augmentations, combinations of the fabricated augmentations? Can you provide further analysis?
>
> **A3:** The generative augmentation is not the combination of fabricated augmentations. As demonstrated in [1, 2], generative augmentations could be “extrapolation” of fabricated augmentations which cannot be achieved through simple combinations. We here add additional analysis of hypergraph generative augmentations. In Figure 4(b), we visualize two hyperedges before and after the generative augmentation, where extraneous information (related to downstream) is removed with hypergraph homophily improved, which is not achievable with (combined) fabricated augmentations (where entities i.e. nodes/hyperedges are perturbed randomly and uniformly).
>
> -----------------------
>
> **Q4:** Eq5: I believe you need a “max” rather than a “min”, as ELBO is to be maximized.
>
> **A4:** Thanks for pointing it out. We’ve revised the text in the paper.
>
> -----------------------
>
> **Q5:** Please explain briefly L_cl as the contrastive learning loss after the corresponding equation in the main text, and say further details are in the Appendix.
>
> **A5:** Thanks for your great suggestion. We’ve revised the text in the paper.
>
> -----------------------
>
> [1] Graph Contrastive Learning Automated. ICML 2021
>
> [2] Bringing Your Own View: Graph Contrastive Learning without Prefabricated Data Augmentations. WSDM 2022

---

### Author Response · Authors · 2022-08-02
**General Response**


We thank all reviewers for thoughtful comments. We would like to respond to the common questions here.


**Q1:** Novelty. (reviewers 9Xc8, RMPZ)

**A1:** Two reviewers are concerned about the technical novelty in this paper versus the conventional graph augmentations. Although sharing the similar philosophy, we respectfully argue that our proposed augmentations are not the direct application from graphs. The fabricated augmentations are specific for higher-order information of hypergraphs in two ways: augmenting on the original view of hypergraph, and on the equivalent view [1] of the transformed bipartite graph. Furthermore, we are the first to propose generative models for hypergraphs with the purpose of augmentation generation, where innovation is also reflected in the generative architecture and optimization design.

[1] You are AllSet: A Multiset Function Framework for Hypergraph Neural Networks. ICLR 2022

-----------------------

**Q2:** Inconclusive experiments of robustness. (reviewers V7YT, 7pvs, RMPZ)

**A2:** Three reviewers are confused on the inconclusive experiments of robustness. We here apologize for the ambiguous presentation and revise accordingly in Sec. 4 Adversarial Robustness. We meant to state (i) we adopt the attack algorithms from the graph domain since there is no existing work for hypergraph attack. (ii) We believe the graph attack algorithms are also applicable to hypergraphs to a certain extent, since both of them are non-Euclidean data structures with sharing properties. (iii) We therefore evaluate our proposed framework in this setting. Importantly, we obtain hypergraph-specific observations, including: (iii.1) HyperGNNs utilizing high-order relations are in general more robust to structure attack compared with GNNs. (iii.2) Accordingly, HyperGCL with hyper-structure augmentation (A2) performs better than with node augmentation (A4). (iii.3) Ultimately, generative augmentation (A6) is capable of surpassing these fabricated augmentations to defend against attacks on all settings. This verifies the robustness of our methods.

-----------------------

**Q3:** Computational cost. (reviewer jPDp, 1bGo)

**A3:** Two reviewers are worried about the potentially expensive computational cost. We here provide the running time comparison in the following two tables (one for dataset statistics and the other for running time) to showcase the time consumption of HyperGCL. Our experimental GPU is Tesla V100 32GB and CPU is Intel(R) Xeon(R) Gold 6240R 2.40GHz. In summary, we observe that HyperGCL is able to be trained with a reasonable time budget (<0.3 seconds for each epoch) and the running time is roughly linearly proportional to the vertex number (dataset size). This indicates our methods can scale up.

|        Stats      |     Cora    |     Pubmed    |     ModelNet40    |
|:-----------------:|:-----------:|:-------------:|:-----------------:|
|       $\|\mathcal{V}\|$     |     2708    |      19717    |        12311      |
|     $\|\mathcal{E}\|$    |     1579    |      7963     |        12311      |
|        $d_v$      |     1.77    |      1.76     |          5        |

|     Epoch/s    |      Cora    |     Pubmed    |     ModelNet40    |
|:--------------:|:------------:|:-------------:|:-----------------:|
|        A0      |     0.016    |      0.102    |        0.078      |
|        A1      |     0.023    |      0.128    |        0.136      |
|        A2      |     0.018    |      0.107    |        0.098      |
|        A3      |     0.026    |      0.133    |        0.107      |
|        A4      |     0.016    |      0.104    |        0.082      |
|        A5      |     0.025    |      0.131    |        0.114      |
|        A6      |     0.034    |      0.189    |        0.296      |

---

> ### Author Response · Authors · 2022-08-08
> **Gentle Reminder**
>
> Dear Reviewers,
>
> Thank you very much for reviewing our paper. We put a lot of effort on rebuttal. We would appreciate if you could check our response to your review comments since the author-reviewer discussion period has started for a few days. This way, if you have further questions and comments, we will still be able to reply before the author-reviewer discussion period ends. If our response resolves your concerns, we kindly ask you to consider raising the rating of our work. Thanks very much for your time and efforts!

---

### Meta-Review · Area_Chair_omoq · 2022-08-26

**Recommendation:** Accept
**Confidence:** Less certain

**Metareview:**

First of all, the majority of reviewers very clearly acknowledged the clarity of presentation and writing within the manuscript. There might not  be a large degree of novelty in the paper, but the approach is well evaluated and exhibits good performance. After carefully reading the author responses to all reviewer comments, my impression is that the authors spent quite some time addressing all raised questions in a satisfactory manner. While the scores of the paper are certainly borderline, I think this paper warrants acceptance which I do recommend at this point.

**Award:**

No

---

### Decision · Program_Chairs · 2022-09-14

Accept